



**Incorporating Recalcitrant Dissolved Organic Carbon and Microbial Carbon Pump Processes into the cGENIE Earth System Model (cGENIEv0.9.35-MCP)**

Yuxian Lai[1], Wentao Ma[2]*, Zihui Zhao[3], Peng Xiu[1], Nianzhi Jiao[1]*

[1]State Key Laboratory of Marine Environmental Science, College of Ocean and Earth Sciences, Fujian

Key Laboratory of Marine Carbon Sequestration, Xiamen University, Xiamen, China

[2]State Key Laboratory of Satellite Ocean Environment Dynamics, Second Institute of Oceanography, MNR, China

[3]Changshan Meteorological Bureau, Quzhou, China

Correspondence to: Wentao Ma (wtma@sio.org.cn), Nianzhi Jiao (jiao@xmu.edu.cn)

**Abstract.** Recalcitrant dissolved organic carbon (RDOC) is a significant component of dissolved organic carbon (DOC), produced through the microbial carbon pump (MCP), and plays a crucial role in long-term carbon sequestration. In this study, we extend the cGENIE Earth System Model by integrating the RDOC fraction and embedding MCP-driven transformations, resulting in the enhanced cGENIE-MCP model.

We implement temperature-dependent limitations on nutrient uptake and organic matter remineralization to simulate MCP processes. Model outputs are compared with contemporary observations and previous cGENIE versions. The model effectively simulates the spatial distribution of concentrations and production rates of labile (LDOC), semi-labile (SLDOC), and RDOC. The cGENIE-MCP model demonstrates improved accuracy over previous versions, capturing spatial variability in DOC pools and

quantifying MCP contributions to long-term carbon sequestration. For instance, sea surface DOC concentrations exhibit a latitudinal gradient, with values ranging from 65-80 μmol kg⁻¹ in tropical-subtropical zones to 40-50 μmol kg⁻¹ in subpolar regions. RDOC concentrations remain relatively stable at 40-50 μmol kg⁻¹ throughout the water column, while LDOC and SLDOC concentrations are typically below 10 μmol kg⁻¹ and 34 μmol kg⁻¹, respectively, in high-production areas. The model reveals a strong

spatial correlation between primary production and LDOC production in upwelling zones, while RDOC production exhibits long-term carbon sequestration. These results emphasize the importance of incorporating MCP processes into Earth system models to better predict ocean carbon sink efficiency and biogeochemical responses to climate change. The cGENIE-MCP model provides a tool for studying the dynamics of ocean DOC and carbon cycle across timescales from paleo to future projections.




## 1 Introduction

The ocean serves as the largest active carbon reservoir on Earth, sequestering nearly 25% of anthropogenic $CO_2$ emissions (Gruber et al., 2023). With rising anthropogenic $CO_2$ emissions, understanding the response of the ocean carbon cycle to future $CO_2$ levels is vital for predicting biogeochemical and physical changes in marine systems (Devries, 2022). Dissolved organic carbon (DOC), generally defined as organic matter passing through 0.2-0.7 µm filters, is the largest ocean reservoir of reduced carbon and plays a key role in ocean biogeochemical cycles (Hansell et al., 2009; Hansell and Carlson, 2014). A significant portion of DOC originates from photosynthesis by phytoplankton in the surface ocean (Hansell, 2013). DOC mainly supports heterotrophic prokaryotes, which respire it to $CO_2$ or transform and diversify it through various metabolic processes (Keller et al., 2014).

DOC can be further categorized based on its degradation rates: labile DOC (LDOC), which degrades within hours to days; semi-labile DOC (SLDOC), with turnover times of weeks to months; and refractory DOC (RDOC), which can persist for tens of thousands of years. LDOC and SLDOC fractions undergo microbial transformations, leading to RDOC production through processes known as the microbial carbon pump (MCP) (Jiao et al., 2010; Jiao et al., 2024b). The MCP is influenced by environmental parameters such as temperature, nutrient availability, and dissolved oxygen concentration (Jiao et al., 2024b; Legendre et al., 2015). It plays a regulatory role in climate change due to RDOC's recalcitrance and large-scale accumulation (Legendre et al., 2015).

The evaluation of the ocean's long-term carbon sequestration potential under climate change relies on numerical models. Despite the advances of the Coupled Model Intercomparison Project Phase 6 (CMIP6), state-of-the-art Earth System Models (ESMs) still exhibit insufficient representation of the MCP and RDOC reservoir (Polimene et al., 2018; Ma et al., 2022). Many ESMs simplify DOC into a single dynamic pool, neglecting the explicit representation of RDOC or the corresponding degradation timescales (Anderson et al., 2015). These simplifications lead to systematic biases: observed RDOC concentrations in deep basins exceed model outputs by up to 40%, while $\Delta^{14}C$ signatures cannot adequately replicate RDOC's millennial-scale lifetime (Yamashita and Tanoue, 2008; Hansell et al., 2012). These discrepancies may arise from the lack of explicit MCP parameterizations that link the transformation of LDOC into RDOC (Hansell et al., 2012; Séférian et al., 2020). Consequently, ESMs underestimate the ocean's carbon sequestration capacity, particularly under global warming scenarios,



where changes in temperature and nutrient availability may alter RDOC accumulation (Séférian et al., 2020; Jokulsdottir, 2011; Sarmiento, 2013; Jiao et al., 2024b; Li et al., 2019).

To address these limitations, integrating MCP processes into ESMs is essential (Ridgwell and Arndt, 2015). While traditional ESMs prioritize high-resolution, near-term climate projections (Séférian et al., 2020), the cGENIE Earth system model emphasizes computational efficiency and flexibility, enabling the exploration of earth system feedbacks over multi-millennial scales (Ridgwell et al., 2007). The GENIE-1 model was initially used by Lenton et al. (2006) to explore the carbon cycle and climate change over millennial timescales. Ridgwell et al. (2007) extended this work to examine the influence of ocean biogeochemistry and marine sediments on atmospheric $CO_2$. The cGENIE model couples ocean biogeochemistry with physical ocean and sea ice dynamics, facilitating studies on ocean carbon sequestration, DOC cycling, and paleoclimate feedbacks (Crichton et al., 2021; Van De Velde et al., 2021; Crichton et al., 2023; Ridgwell and Zeebe, 2005; Anagnostou et al., 2016; Boscolo-Galazzo et al., 2021; Vervoort et al., 2024; Ma and Tian, 2014; Wang et al., 2023; Ridgwell et al., 2007).

Despite recent advances, the explicit representation of MCP processes and RDOC cycling within cGENIE has remained limited. The Minnesota Earth System Model for Ocean biogeochemistry (MESMO 3), an Earth system model of intermediate complexit derived from cGENIE, includes an explicit treatment of semi-labile and refractory DOM pools (Matsumoto et al., 2021). However, MESMO 3 lacks a mechanistic representation of DOM production pathways associated with MCP processes—specifically, the transformation of semi-labile DOC (SLDOC) into RDOC—and has not been calibrated against global DOM observations. In this study, we propose an extension to the cGENIE model that integrates RDOC and MCP processes to investigate their long-term response to climate change. We introduce a new framework, cGENIE-MCP, which partitions total DOC into three distinct fractions—labile (LDOC), semi-labile (SLDOC), and refractory (RDOC)—enabling improved simulation of DOC production and remineralization based on prior formulations from cGENIE and MESMO 3. We evaluate the performance of the cGENIE-MCP model against global observational datasets and compare its outputs with those from the standard cGENIE model. Finally, we analyze the spatial distribution and production of LDOC, SLDOC, and RDOC in relation to primary production to assess the model's capability in capturing essential features of the MCP.

**2 Model description**

**2.1 The cGENIE model framework**



The cGENIE model, derived from the Grid-Enabled Integrated Earth System Model (GENIE), is an Earth system model of intermediate complexity (EMIC) designed for simulating long-term carbon cycle dynamics. It is structured based on a 3-D frictional geostrophic ocean model, a 2-D energy-moisture balance model (EMBM), and a dynamic-thermodynamic sea-ice model, all within a modular framework that integrates essential Earth system components. These include ocean circulation, marine biogeochemistry, ocean-atmosphere and ocean-sediment exchanges, and the long-term geological carbon cycle on land (Lenton et al., 2006; Van De Velde et al., 2021; Marsh et al., 2011; Williamson et al., 2006; Ridgwell). Compared to high-resolution Earth System Models (ESMs), cGENIE operates at coarser spatial and temporal resolutions, offering enhanced computational efficiency and flexibility. This makes it particularly suitable for exploring Earth system feedbacks over multi-millennial timescales.

In this study, we adopt the global ocean circulation configuration of Cao et al. (2009), which forms the basis for numerous subsequent biogeochemical developments within cGENIE (Crichton et al., 2021; Reinhard et al., 2020; Reinhard et al.). Physical parameters and boundary conditions governing the climate system are retained from Cao et al. (2009). The model includes an explicit representation of iron cycling and co-limitation of primary productivity, following the formulation by Ward et al. (2018). Simulations are initialized from a pre-industrial climate state with atmospheric $CO_2$ set at 278 ppm. The model is configured on a 36×36 horizontal grid with equal area (i.e. equal divisions in longitude and the sine of latitude) and 16 vertical ocean layers of increasing thickness (Ridgwell et al., 2007; Anagnostou et al., 2016).

## 2.2 Ocean biogeochemical module

The ocean biogeochemistry in cGENIE is simulated using the BIOGEM module, which models nutrient-driven biological productivity and the cycling of carbon and associated elements (Van De Velde et al., 2021). BIOGEM includes representations of air-sea gas exchange, nutrient uptake by primary producers, and the remineralization of organic matter in the water column. Phytoplankton are not explicitly represented; instead, primary productivity is calculated diagnostically based on the availability of limiting nutrients (phosphate and dissolved iron), solar radiation, and temperature, following previous studies (Ward et al., 2018; Matsumoto et al., 2008b; Tanioka et al., 2021; Matsumoto et al., 2008a; Matsumoto et al., 2021). Phosphate ($PO_4$) and other nutrients are converted to particulate organic matters (POMs) in the euphotic zone according to the Redfield stoichiometry. POMs are exported from the surface layer and undergo depth-dependent remineralization, releasing nutrients and carbon back into the water column. In this study, we extend the BIOGEM module by partitioning DOC into three fractions—LDOC,





SLDOC, and RDOC—to better represent MCP processes. A fraction of primary production is allocated

to each DOC pool. LDOC degradation is regulated by temperature, SLDOC degradation is driven

primarily by photodegradation, and a portion of SLDOC is converted into RDOC. The transformation

from SLDOC to RDOC is governed by the characteristic lifetime of SLDOC, reflecting microbial

reprocessing pathways consistent with the MCP framework. The updated DOC cycling scheme enables

us to track the production, transformation, and persistence of different DOC fractions under varying

environmental conditions. In the following sections, we provide a detailed description of these carbon

cycling processes and the associated parameterizations.

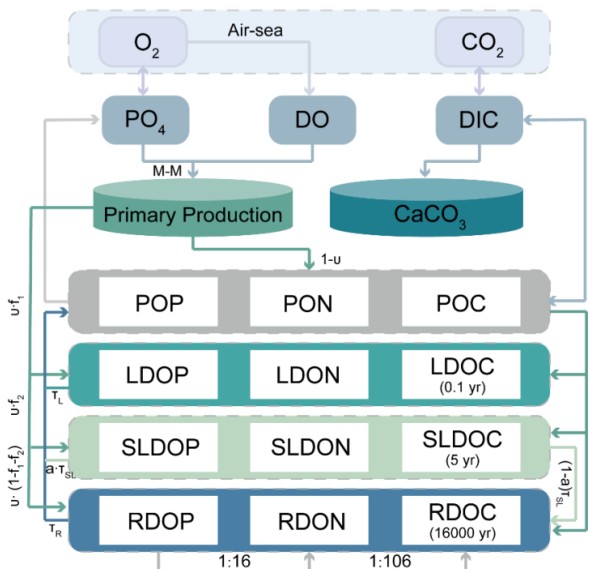

Figure1. Schematic representation of phosphorus, carbon, and oxygen cycling in the BIOGEM module

of the cGENIE-MCP model.

### 2.2.1 Ocean-atmosphere gas exchange

The flux of $CO_2$ across the air-sea interface is calculated following Ridgwell et al. (2007):

$$f_{atm \to ocn}^{CO_2} = k \cdot Sol \cdot (pCO_{2atm} - pCO_{2sea}),$$

where $pCO_{2atm}$ and $pCO_{2sea}$ are the partial pressure of $CO_2$ (µatm) in the atmosphere and ocean

surface, respectively. $k$ is the gas transfer velocity (m s$^{-1}$), and $Sol$ (mol m$^{-3}$ µatm$^{-1}$) is solubility of $CO_2$

in sea water. The gas transfer velocity is calculated as:



$$k = l \cdot [2.5 \cdot (a + b \cdot T + c \cdot T^2) + 0.38 \cdot \mu_{av}^2] \cdot \left(\frac{Sc}{660}\right)^{-0.5},$$

where $a$, $b$, and $c$ are constants, with values of 0.5246, 0.01625, and 4.9946×10$^{-4}$, respectively; T is the ocean temperature in Kelvin (K). $l$ is the scaling factor; $\mu_{av}$ is the average wind speed (m s$^{-1}$); and Sc is the Schmidt number, calculated as:

$$Sc = A - B \cdot T + C \cdot T^2 - D \cdot T^3,$$

where $A$, $B$, $C$, and $D$ are empirical constants with constants: A=2073.1, B=125.62, C=3.6267, and D=0.0432.

The total CO₂ flux is area-weighted by the ice-free ocean surface area:

$$F_{ocn \to atm}^{CO_2} = \sum_c (f_{atm \to ocn}^{CO_2} \cdot (A_{(c)} - A_{ice(c)})),$$

where $A_{(c)}$ is the grid cell area and $A_{ice(c)}$ is the ice-covered portion of the cell.

### 2.2.2 Biological production

Nutrient uptake in each surface grid cell is given by:

$$\Gamma = (1 - A) \cdot \gamma^T \cdot \min\left(\frac{PO_4^{3-}}{PO_4^{3-} + K^{PO_4^{3-}}}, \frac{FeT}{FeT + K^{FeT}}\right) \cdot \frac{I}{I_0} \cdot \frac{\min(PO_4^{3-}, red_{P/Fe} \cdot FeT)}{\tau_{bio}},$$

$$\gamma^T = ae^{\left(\frac{T}{b}\right)},$$

Where $A$ represents the ice-covered fraction of the grid cell, the temperature factor $\gamma^T$ is derived from Eppley's initial values, with a = 0.59 and b = 75.80, which is associated with the Q₁₀ factor. $K^{PO_4^{3-}}$ and $K^{FeT}$ are half-saturation constants for phosphate and iron (mol kg$^{-1}$), respectively. $PO_4^{3-}$ is the phosphate concentration (mol kg$^{-1}$), and $FeT$ is the iron concentration (mol kg$^{-1}$). $I$ is the local solar radiation, and $I_0$ is the solar constant, $red_{P/Fe}$ is the Fe:P Redfield ratio, $\tau_{bio}$ is an optimal uptake timescale (h).

The primary production ($F_{z=h_e}^{POP}$) is derived from nutrient uptake (mol m$^{-2}$ yr$^{-1}$). Biologically assimilated PO₄ is converted into particulate organic phosphorus (POP), with a fraction partitioned into labile dissolved organic phosphorus (LDOP), semi-labile dissolved organic phosphorus (SLDOP), and refractory dissolved organic phosphorus (RDOP). The total phosphorus uptake is partitioned as:

$$\int_{h_e}^{0} \rho \cdot \Gamma \, dz = F_{z=h_e}^{POP} + F_{z=h_e}^{LDOP} + F_{z=h_e}^{SLDOP} + F_{z=h_e}^{RDOP},$$

with each term defined as:

$$F_{z=h_e}^{POP} = \int_{h_e}^{0} \rho \cdot (1 - v) \cdot \Gamma \, dz,$$





$$F_{z=h_e}^{LDOP} = \int_{h_e}^{0} \rho \cdot v \cdot f_1 \cdot \Gamma \, dz,$$

$$F_{z=h_e}^{SLDOP} = \int_{h_e}^{0} \rho \cdot v \cdot f_2 \cdot \Gamma \, dz,$$

$$F_{z=h_e}^{RDOP} = \int_{h_e}^{0} \rho \cdot v \cdot (1 - f_1 - f_2) \cdot \Gamma \, dz,$$

where $v = 0.66$ is the fraction of nutrient uptake allocated to dissolved organic phosphorus following

the OCMIP-2 protocol (Ridgwell et al., 2007), $f_1$=0.9599 and $f_2$=0.04 are the export partitioning

coefficients for LDOP and SLDOP, respectively, based on Wang et al. (2023), $\rho$ is the seawater density

(kg m$^{-3}$), $\Gamma$ is the nutrient uptake rate, and $h_e$ is the depth of the euphotic layer (m). Organic phosphorus

fluxes are converted to organic carbon fluxes using the Redfield ratio (C:N:P=106:16:1):

$$F_{z=h_e}^{POC} = 106 \cdot F_{z=h_e}^{POP},$$

$$F_{z=h_e}^{LDOC} = 106 \cdot F_{z=h_e}^{LDOP},$$

$$F_{z=h_e}^{SLDOC} = 106 \cdot F_{z=h_e}^{SLDOP},$$

$$F_{z=h_e}^{RDOC} = 106 \cdot F_{z=h_e}^{RDOP}.$$

POC is categorized to labile POC (POC1) and recalcitrant POC (POC2) based on their relative

degradability, following Ridgwell et al. (2007):

$$F_{z=h_e}^{POC1} = 0.992 \cdot F_{z=h_e}^{POC},$$

$$F_{z=h_e}^{POC2} = 0.008 \cdot F_{z=h_e}^{POC}.$$

This implies that 0.8% of the total POC flux is assigned to the recalcitrant pool, with the remaining 99.2%

attributed to the labile pool.

### 2.2.3 Remineralization in the water column

Remineralization in the water column converts portions of POC and DOC into inorganic nutrients,

supporting nutrient cycling. While some POC sinks to the bottom waters and is eventually buried in

sediments, a substantial fraction is remineralized within the water column.

**POC remineralization**

The remineralization of POC is modeled as a temperature-dependent process, with separate

treatment for labile (POC1) and recalcitrant (POC2) components. The remineralization rate is given by:

$$k_{zPOCn} = \beta_{POCn} \cdot e^{\left(\frac{-Ea_n}{RT_z}\right)},$$





$$F_{rz}^{POCn} = k_{zPOCn} \cdot POCn_z,$$

where $POCn_z$ is the concentration of POC component n (either POC1 or POC2) at depth z; $k_{zPOCn}$ represents the remineralization rate of $POC_n$; $E_{an}$ is the activation energy of POC component n

(J mol$^{-1}$); R is the gas constant (J K$^{-1}$ mol$^{-1}$); $T_z$ is the temperature at z layer (K); $\beta_{POC_n}$ is the remineralization rate constant for $POC_n$ (yr$^{-1}$). $F_{rz}^{POCn}$ denotes the remineralization flux of $POC_n$ at z layer.

**LDOC remineralization**

LDOC is rapidly remineralized in the water column, releasing inorganic carbon and nutrients. The

remineralization follows a similar temperature-dependent formulation:

$$k_{zLDOC} = \beta_{LDOC} \cdot e^{\left(\frac{-E_{aL}}{RT_z}\right)},$$

$$F_{rz}^{LDOC} = k_{zLDOC} \cdot LDOC_z,$$

where $k_{zLDOC}$ is the temperature-dependent remineralization rate of LDOC at z layer; $E_a$ is the activation energy (J mol$^{-1}$); R is the gas constant (J K$^{-1}$ mol$^{-1}$); $T_z$ is the temperature at z (K); $\beta_{LDOC}$ is

the remineralization rate constant (yr$^{-1}$); $LDOC_z$ is the LDOC concentration at z layer (mol kg$^{-1}$). $F_z^{LDOC}$ denotes the remineralization flux of LDOC at z layer.

**SLDOC remineralization**

SLDOC remineralizes more slowly than LDOC and contributes to nutrient cycling on intermediate timescales. The remineralization rate is described by:


$$k_{zSLDOC} = \frac{1}{\tau_{SL}},$$

$$F_{rz}^{SLDOC} = k_{zSLDOC} \cdot SLDOC_z,$$

where $k_{zSLDOC}$ is the temperature-dependent remineralization rate of SLDOC at z layer (yr$^{-1}$); $\tau_{SL}$ is the SLDOC lifetime (yr); $SLDOCz$ is the SLDOC concentration at z layer (mol kg$^{-1}$). $F_z^{SLDOC}$ denotes the remineralization flux of SLDOC at z layer.

**RDOC remineralization**

RDOC remineralizes on much longer timescales and is partly derived from the transformation of SLDOC. The RDOC remineralization and conversion fluxes are:

$$k_{zRDOC} = \frac{1}{\tau_R},$$

$$F_{conv}^{RDOC} = F_{rz}^{SLDOC} \cdot a,$$


$$F_{rz}^{RDOC} = k_{zRDOC} \cdot RDOC_z,$$





where $k_{zRDOC}$ is the temperature-dependent remineralization rate of RDOC at z layer (yr⁻¹); $\tau_R$ is the RDOC lifetime (yr); $RDOCz$ is the RDOC concentration at z layer. $F_{conv}^{RDOC}$ denotes the remineralization flux of converted from SLDOC to RDOC at z layer. $a$ is the conversion rate from SLDOC to RDOC, which is based on the Wang et al. (2023); $F_z^{RDOC}$ denotes the remineralization flux of RDOC at z layer.


**Table 1. Parameters of biogeochemical processes in the cGENIE-MCP model**

| Parameters | Symbol | Value | Unit | Reference |
|---|---|---|---|---|
| The scaling factor | $l$ | $2.778×10^{-6}$ | - | (Crichton et al., 2021) |
| The half-saturation constant | $K^{PO_4^{3-}}$ | $0.05×10^{-6}$ | mol kg⁻¹ | (Crichton et al., 2021) |
| | $K^{FeT}$ | $0.03×10^{-9}$ | mol kg⁻¹ | (Matsumoto et al., 2021) |
| A proportion of PO₄ take up by the biota is partitioned into DOP | $\nu$ | 0.66 | - | (Najjar et al., 1992; Orr et al., 2017) |
| Optimal uptake timescale | $\tau_{bio}$ | 95.63 | h | (Van De Velde et al., 2021) |
| A proportion of PO₄ take up by the biota is partitioned into LDOP | $f_1$ | 0.9599 | - | (Wang et al., 2023) |
| A proportion of PO₄ take up by the biota is partitioned into SLDOP | $f_2$ | 0.04 | - | (Wang et al., 2023) |
| The activation energy of POC₁ | $E_{a1}$ | 54000 | J mol⁻¹ | (Crichton et al., 2021) |
| The activation energy of POC₂ | $E_{a2}$ | 80000 | J mol⁻¹ | (Crichton et al., 2021) |
| The activation energy of LDOC | $E_{aL}$ | 54000 | J mol⁻¹ | (Crichton et al., 2021) |
| The SLDOC lifetime | $\tau_{SL}$ | 5 | yr | (Letscher et al., 2015) |
| The RDOC lifetime | $\tau_R$ | 16000 | yr | (Letscher et al., 2015) |
| A proportion of SLDOC take up by the biota is partitioned into RDOC | $a$ | 0.015 | - | (Wang et al., 2023) |

## 2.3 Model validation method



To evaluate the performance of the cGENIE-MCP model, we compare simulated outputs with climatological and observational datasets. Modeled ocean temperature, salinity, PO₄, and DO are evaluated against the World Ocean Atlas 2023 (WOA23) climatology. DIC and alkalinity are compared with the Global Ocean Data Analysis Project (GLODAP) dataset (Key et al., 2004; Lauvset et al., 2024; Lauvset et al., 2021; Lauvset et al., 2023). Model skill is quantified using the root mean square deviation

(RMSE):

$$RMSE = \sqrt{\frac{1}{N}\sum_{i=1}^{N}[(X_i - \bar{X}) - (Y_i - \bar{Y})]^2},$$

where $X_i$ and $\bar{X}$ represent the simulated tracer concentration at position i and its mean across time, $Y_i$ and $\bar{Y}$ denote the observed tracer concentration and its mean; $N$ is the number of data points used in the calculation.

For consistency, WOA23 and GLODAP datasets are interpolated onto the cGENIE-MCP model grid (36×36×16 grids). Ocean regions used for the validation of vertical profiles are based on regional divisions from Crichton et al. (2021) with minor modifications to ensure each region maintains consistent oceanographic characteristics and particle flux regimes (Figure 2).

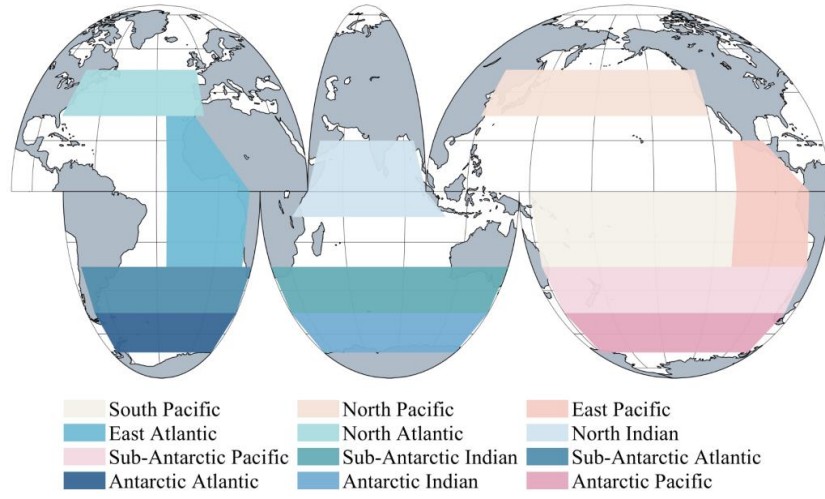

Figure 2. Ocean regions used for model validation in the cGENIE-MCP simulations. Colored regions indicate the spatial domains included in the validation of water column biogeochemical properties. Unshaded areas are excluded from the analysis due to limited relevance or observational data availability.

**3 Model Results**

**3.1 Temperature**



Ocean temperature strongly influences biogeochemical cycling and nutrient uptake, as demonstrated in prior studies (Yan et al., 2024). To assess the performance of the cGENIE-MCP model, simulated temperature fields were evaluated against observations from the WOA23 at three representative depth

levels: surface, intermediate (400 m), and deep (3000 m) layers. As shown in Fig. 3, the cGENIE-MCP model reproduces the large-scale spatial distribution of ocean temperature with good accuracy, especially in the surface layer and in low-latitude regions. Temperature exhibits a clear latitudinal gradient, decreasing from the tropics toward the poles in the upper and intermediate ocean, with relatively small variability in the deep ocean (Fig. 4). The RMSE in the tropical regions is approximately 1.2 °C, while

RMSE values in the Atlantic Ocean range from 0.6 °C to 1.0 °C, indicating strong agreement with observations (Table 2).

Despite the model performing well overall, it underestimates temperature in high-latitude regions, deep layers, and equatorial upwelling zones. These biases likely arise from the limitations of the simplified two-dimensional energy-moisture balance atmospheric model used in cGENIE, which affects

surface forcing and ocean circulation. Additionally, insufficient stratification in the upper ocean and uncertainties associated with the interpolation of WOA23 data onto the model grid may also contribute to discrepancies in the water column.

Comparisons between the standard cGENIE and cGENIE-MCP versions reveal negligible differences in temperature fields, with RMSE values of 0.01 °C (Fig. 3-4 and Table 2). These results

confirm that incorporating the refractory dissolved organic matter (RDOM) component into the cGENIE-MCP model does not degrade the model's performance in simulating ocean temperature, thereby preserving the core physical integrity of the system.

**Table 2. RMSE of modeled tracers for cGENIE-MCP and cGENIE compared to observational**

**data**

| Tracers | cGENIE-MCP | cGENIE |
|---------|------------|--------|
| T (°C) | 1.25 | 1.26 |
| Salinity | 0.34 | 0.34 |
| $PO_4$ ($\mu$mol kg$^{-1}$) | 0.10 | 0.12 |
| DO ($\mu$mol kg$^{-1}$) | 26.89 | 28.50 |
| DIC ($\mu$mol kg$^{-1}$) | 51.33 | 50.92 |



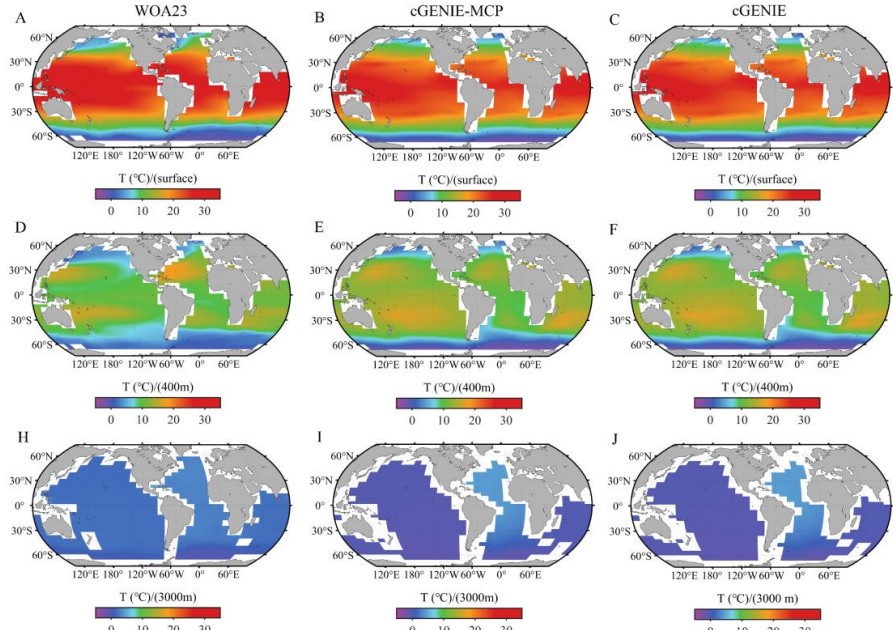

Figure 3. Ocean temperature distributions (°C) at three depth levels: (A-C) the surface, (D-F) 400 m, and (H-J) 3000 m. Panels show: (A, D, H) observational data from WOA23; (B, E, I) simulated results from the cGENIE-MCP model; and (C, F, J) outputs from the standard cGENIE.






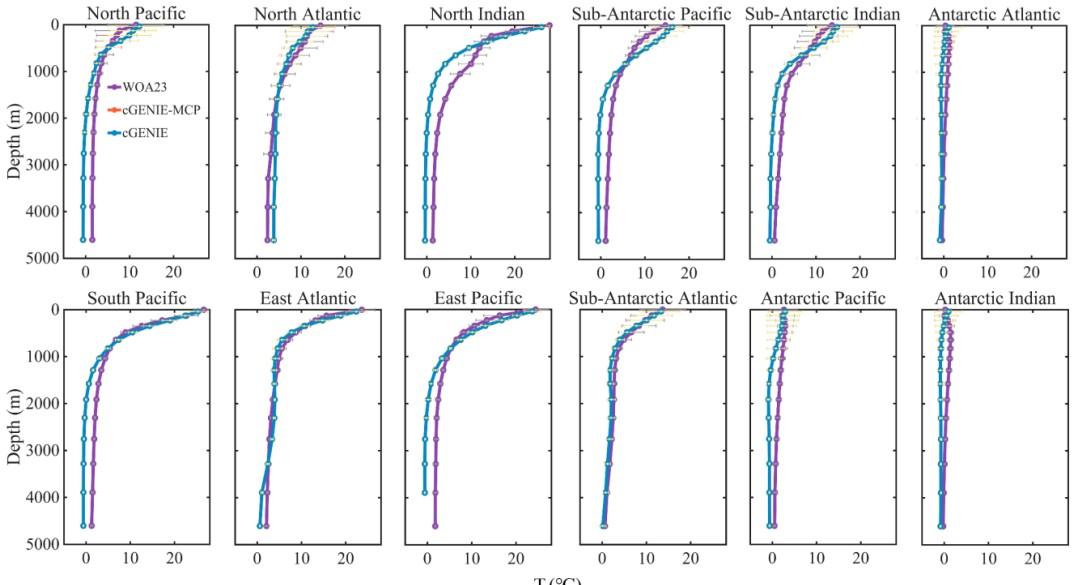

Figure 4. Vertical profiles of temperature(°C) in selected ocean regions, comparing among WOA23
data, cGENIE-MCP and standard cGENIE model outputs. Regions include the Atlantic Ocean, low-
latitude tropics, South Pacific, and Northern Indian Ocean, as defined in the model evaluation.

**3.2 Phosphate**

Fig. 5 illustrates the spatial distribution of $PO_4$ concentration in both surface and deep water. Both
the cGENIE-MCP and standard cGENIE models show good agreement with WOA23 dataset. Modeled
$PO_4$ concentrations range from near 0 to a maximum of 4.5 μmol kg$^{-1}$, with lower values observed in the
surface layer and upper thermocline. In surface waters of low- and mid-latitude regions, concentrations
vary from 0 to 0.5 μmol kg$^{-1}$, exhibiting an increasing trend toward higher latitudes, consistent with prior
studies (Fig. 5).

Fig. 6 shows regional vertical profiles of $PO_4$ concentrations, comparing WOA23 data with outputs
from both the cGENIE-MCP and standard cGENIE models. The cGENIE-MCP model reproduces
vertical structures more accurately in middle- and high-latitude regions, notably in the sub-Antarctic zone,
Northwest Pacific, and Eastern Pacific, where observed $PO_4$ concentrations peak within the thermocline.
In these regions, the cGENIE-MCP achieves a RMSE of approximately 0.1 μmol kg$^{-1}$. In the deep ocean,
$PO_4$ concentrations are lowest in the North Atlantic and highest in the North Pacific, with intermediate
levels in the South Indian and Southern Oceans.

Some discrepancies are observed in the intermediate and deep layers of the Indian, Atlantic, and





North Pacific Oceans, where both models tend to overestimate PO₄ concentrations compared to observations. These discrepancies are consistent with those reported in previous studies (Stappard et al.,

2025). The overall RMSE between cGENIE-MCP and WOA23 is 0.1 μmol kg⁻¹, indicating that the model is capable of reproducing large-scale phosphate distributions with a high degree of accuracy.

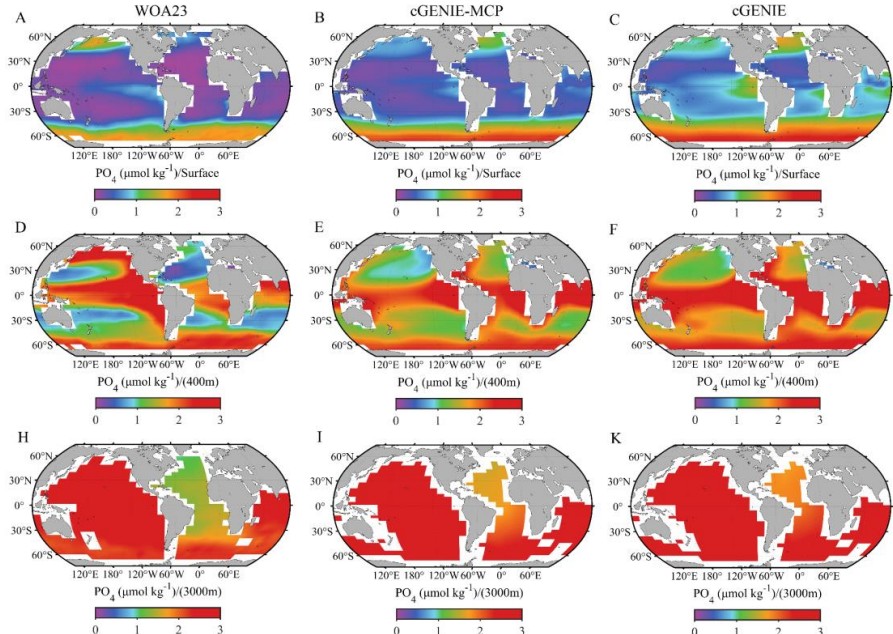

Figure 5. Spatial distributions of PO₄ concentrations (μmol kg⁻¹) at three depth levels: (A-C) surface, and (D-F) 400 m, and (H-J) 3000 m. Panels show: (A, D, H) observational data fromWOA23; (B, E, I) simulated results from the cGENIE-MCP model; and (C, F, J) outputs from the standard cGENIE (μmol kg⁻¹).



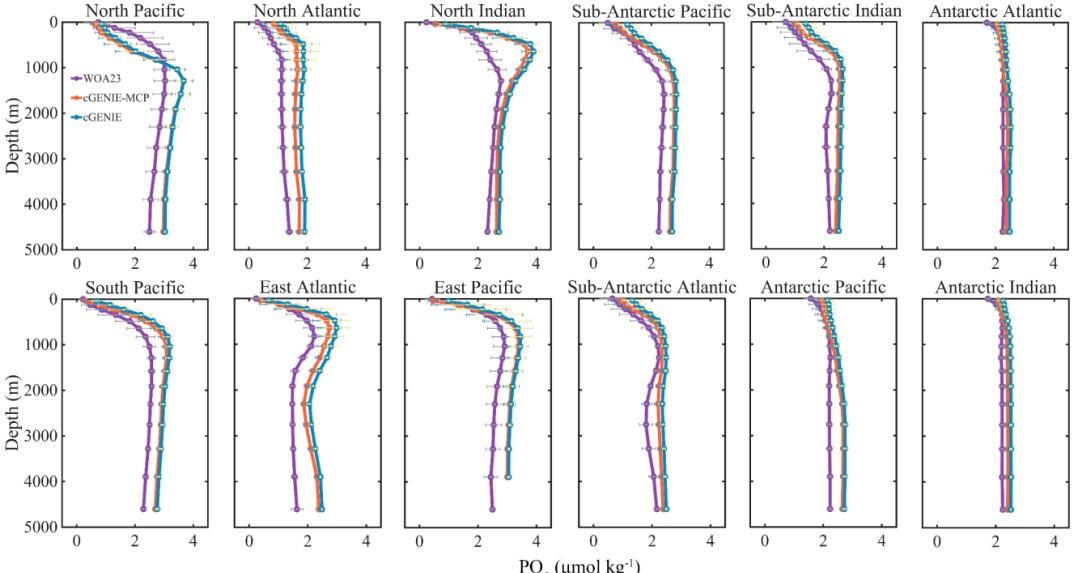

Figure 6. Vertical profiles of PO₄ ($\mu$mol kg$^{-1}$) in selected ocean regions, comparing among WOA23 data, cGENIE-MCP and standard cGENIE model outputs.

### 3.3 Dissolved oxygen

DO is a critical determinant of aerobic respiration and species distribution in marine ecosystems. Figure 7 shows that the simulated spatial distribution of DO aligns well with the WOA23 dataset, with an RMSE of 50-52 $\mu$mol kg$^{-1}$. The spatial pattern of DO varies significantly in both horizontal and vertical dimensions (Alhassan et al., 2024). DO concentrations decline with depth, reaching minima within the oxygen minimum zones (OMZs) typically located between 200 and 1000 m. The cGENIE-MCP model successfully reproduces these vertical and horizontal DO gradients, particularly in the Atlantic Ocean (Fig. 8). However, moderate discrepancies are observed in the Antarctic region, potentially reflecting limitations in the simulation of regional ocean circulation.



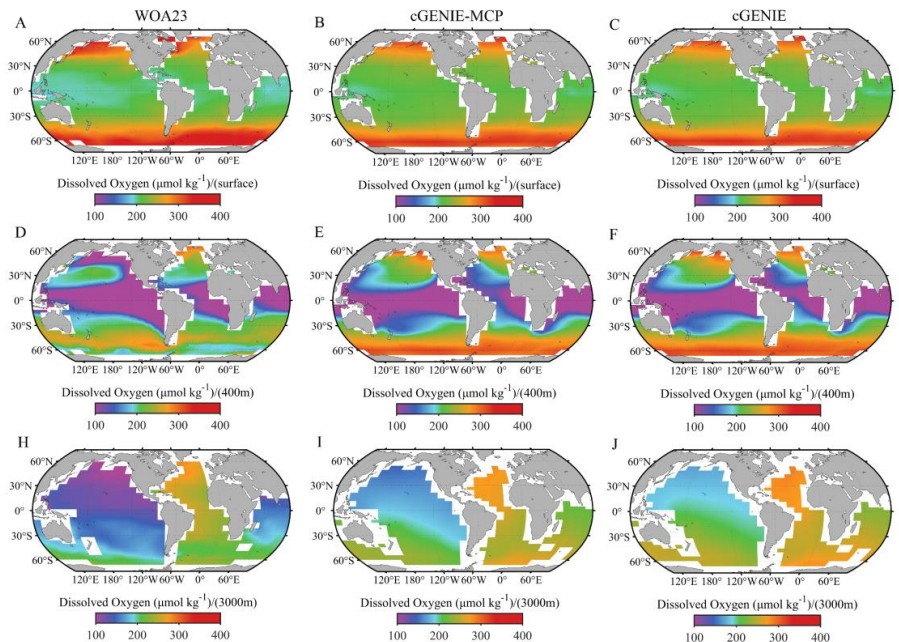

Figure 7. Spatial distributions of DO concentrations (μmol kg⁻¹) at three depth levels: (A-C) surface,

and (D-F) 400 m, and (H-J) 3000 m. Panels show: (A, D, H) observational data fromWOA23; (B, E, I)

simulated results from the cGENIE-MCP model; and (C, F, J) outputs from the standard cGENIE (μmol

kg⁻¹).

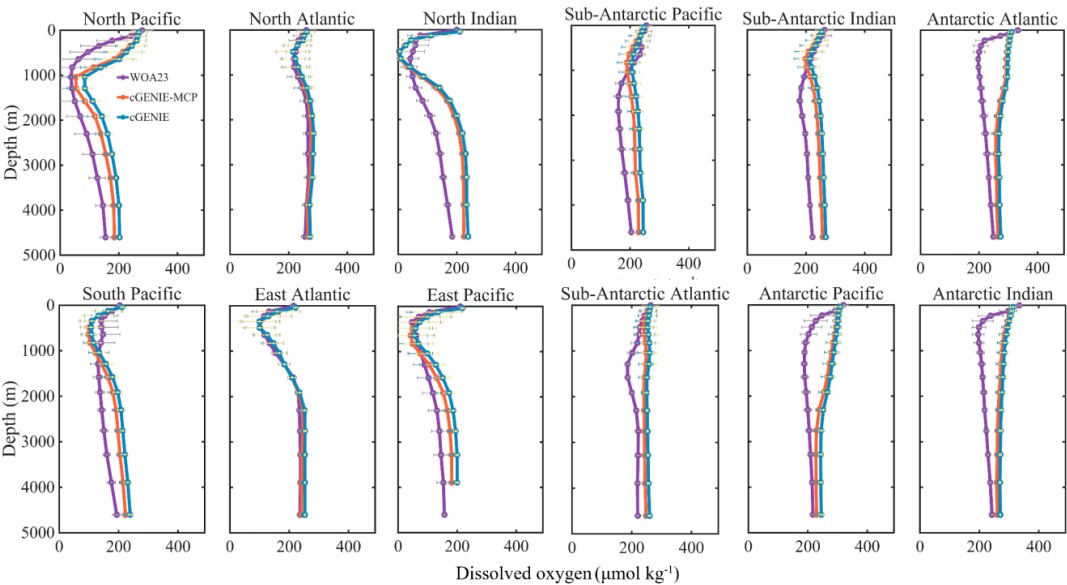



Figure 8. Vertical profiles of DO (µmol kg$^{-1}$) in selected ocean regions, comparing among WOA23 data, cGENIE-MCP and standard cGENIE model outputs.

### 3.4 Dissolved inorganic carbon

The global distribution of surface dissolved inorganic carbon (DIC) serves as a key indicator of oceanic carbon cycling, shaped by atmospheric $CO_2$ levels, alkalinity, and biological activity. Figures 9-10 compare the simulated surface DIC concentrations from the cGENIE-MCP model with observations from the GLODAP dataset. The model generally underestimates DIC concentrations relative to observations, primarily due to the use of pre-industrial atmospheric $CO_2$ levels (278 ppm), whereas present-day $CO_2$ concentrations are substantially higher. When the model is forced with modern $CO_2$ levels (400 ppm), the agreement with the GLODAP dataset improves significantly (Fig. S1). Additional factors contributing to the discrepancy include uncertainties in surface alkalinity, biological carbon uptake, and remineralization-driven carbon release (Wu et al., 2019).

Surface and subsurface alkalinity distributions are also evaluated (Figs. S2-S3). Similar to DIC, simulated alkalinity concentrations are lower than observed values, reflecting consistent biases across carbon system variables. The global pattern of surface DIC (Fig. 9) closely mirrors that of major nutrients such as phosphate ($PO_4^{3-}$), with elevated concentrations in high-latitude regions and reduced levels in low-latitude zones. This distribution highlights the role of temperature and physical dynamics— particularly vertical mixing—in shaping surface DIC concentrations. Lower temperature and well-mixed high-latitude surface waters typically support higher DIC levels, whereas warmer, more stratified low-latitude waters exhibit lower concentrations.





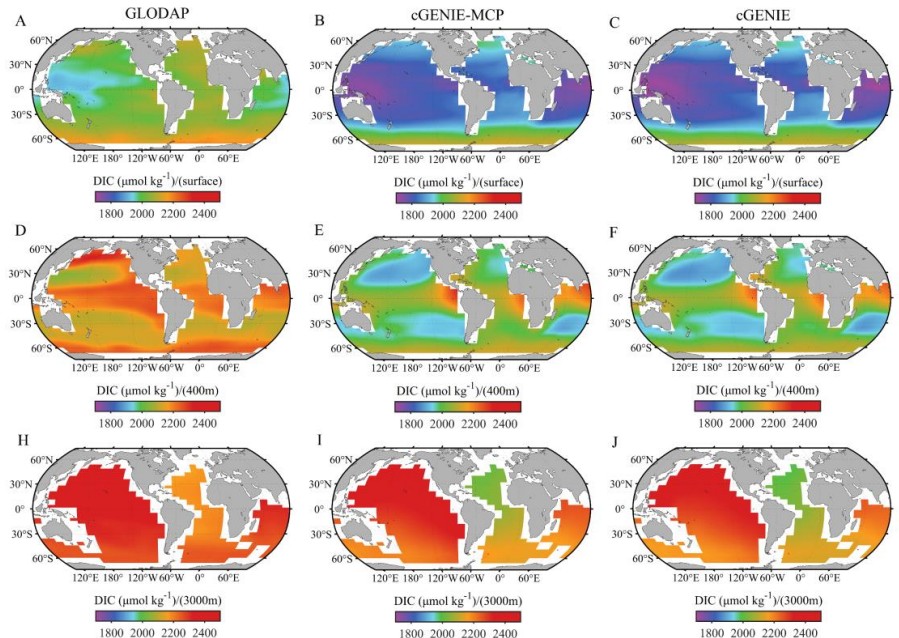

Figure 9. Spatial distributions of DIC concentrations (µmol kg⁻¹) at three depth levels: (A-C) surface, and (D-F) 400 m, and (H-J) 3000 m. Panels show: (A, D, H) observational data from GLODAP; (B, E, I) simulated results from the cGENIE-MCP model; and (C, F, J) outputs from the standard cGENIE (µmol kg⁻¹).

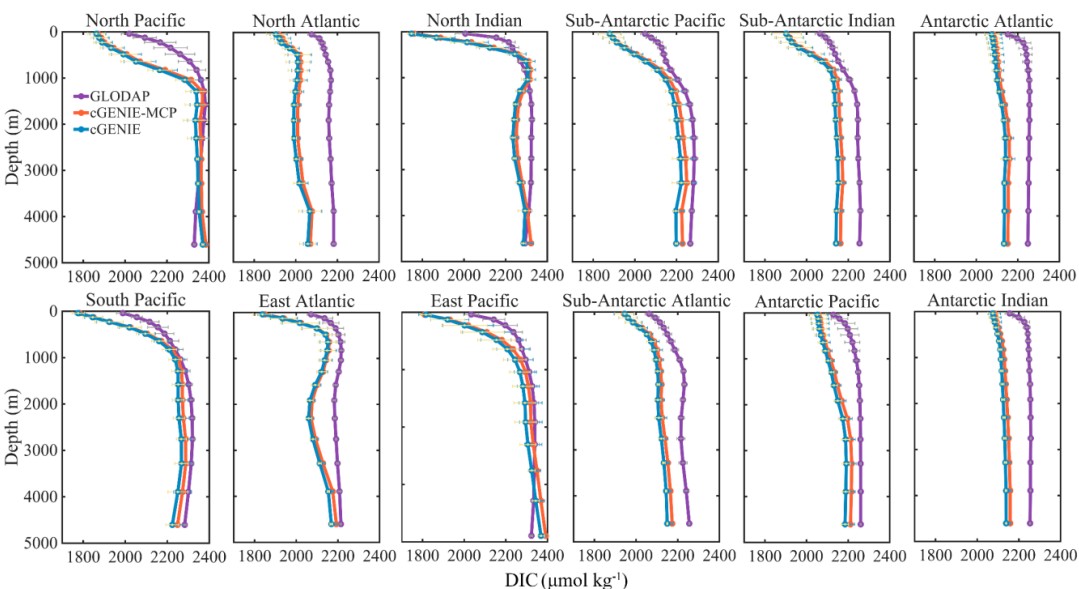



Figure 10. Vertical profiles of DIC (μmol kg⁻¹) in selected ocean regions, comparing among GLODAP data, cGENIE-MCP and standard cGENIE model outputs.

**3.5 Dissolved organic carbon**

DOC is a key component of ocean carbon cycling, shaped by biological production, ventilation, and circulation dynamics. Figure 11 illustrates the global distribution of surface DOC concentrations simulated by the cGENIE-MCP model and compares them with observational data. Simulated surface DOC concentrations range from 40 to 80 μmol kg⁻¹, with elevated values (70-80 μmol kg⁻¹) occurring

primarily in tropical and subtropical regions between ~40°N and 40°S. These elevated concentrations reflect the combined effects of high biological production and strong vertical stratification, which limits vertical mixing and facilitates DOC accumulation in surface waters. Compared to MESMO 3—which substantially overestimates DOC concentrations, with surface values reaching 70–150 μmol kg⁻¹ and deep values of 60–70 μmol kg⁻¹—the cGENIE-MCP model exhibits improved agreement with observed DOC

distributions in both surface and deep layers. However, the model underrepresents equatorial upwelling, consistent with similar limitations in simulating temperature fields, leading to discrepancies in equatorial DOC concentrations. In contrast, lower surface DOC concentrations (40-50 μmol kg⁻¹) are simulated in subpolar regions and the Southern Ocean (Fig. 12), consistent with enhanced vertical mixing and the upwelling of deep DOC-poor waters (Carlson et al., 2010). In additionally, Hansell (2013) studies have

indicated that in the western Sargasso Sea region, the DOC concentration at 150-200 meters is approximately 56 μmol kg⁻¹, the SLDOC usually reaches approximately 11 μmol kg⁻¹, and the RDOC concentration is approximately 46 μmol kg⁻¹, which is consistent with our simulation results (Fig. S6A). The large-scale patterns of simulated DOC distributions, such as elevated values in stratified subtropical gyres and lower concentrations in high-latitude and upwelling regions, are qualitatively consistent with

observations (Hansell et al., 2012; Carlson et al., 2010; Hansell, 2013), which suggest that the patterns of DOC cycling are approximately reasonable.

LDOC is rapidly remineralized and does not accumulate significantly in surface waters, with concentrations typically below 10 μmol kg⁻¹ in regions of high productivity. SLDOC concentrations range from 10 to 30 μmol kg⁻¹ in tropical and subtropical regions, peaking at approximately 34 μmol kg⁻¹,

indicative of high production efficiency in these zones (Carlson et al., 2010; Hansell and Carlson, 2014; Hansell, 2013). RDOC concentrations range from 40 to 50 μmol kg⁻¹ in the upper ocean, with maximum values found in the centers of subtropical gyres. These high concentrations are associated with convergent circulation and weak vertical mixing, which facilitate RDOC accumulation. The RDOC concentration





and its vertical variation simulated by the model are in agreement with the results of Wang et al. (2023)
(Figure S6B). The simulation of LDOC, SLDOC, and RDOC captures the essential patterns consistently
reported in observational and previous studies (Table S2) (Letscher et al., 2015; Legendre et al., 2015;
Hansell and Carlson, 2014; Hansell, 2013; Hansell et al., 2012; Hansell et al., 2009; Ge et al., 2022).
Specifically, the model reproduces the rapid turnover and low accumulation of labile fractions in
productive regions.

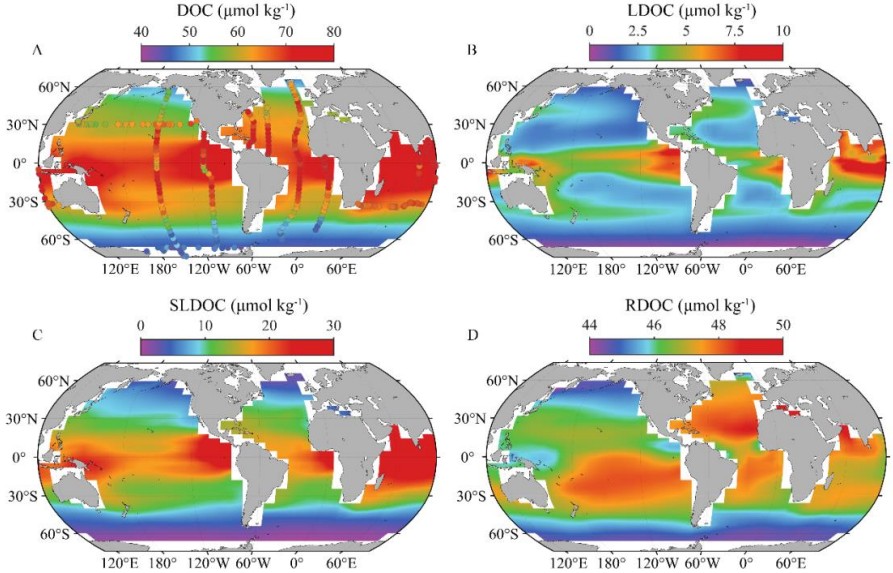

Figure 11. Global distributions of surface (A) DOC, (B)LDOC, (C)SLDOC, (D)RDOC concentrations
(µmol kg⁻¹). Shaded represents cGENIE-MCP model outputs, overlaid with scatter points of observed
values from Hansell's laboratory (Hansell, 2013).

Figure 12 shows the vertical distribution of modeled DOC concentrations compared with
observational data. In the euphotic zone, mean DOC concentrations exceed 60 µmol kg⁻¹, with maxima
over 70 µmol kg⁻¹ observed in the Atlantic, Pacific, and Indian Oceans. Within the mesopelagic and
bathypelagic zones, DOC concentrations range from 40 to 60 µmol kg⁻¹, with smoother vertical gradients
at high latitudes due to reduced remineralization rates, strong convective mixing, and enhanced isopycnal
ventilation.

In the North Atlantic, the formation of North Atlantic Deep Water (NADW) facilitates the transport
of DOC to depth, with concentrations reaching up to 47 µmol kg⁻¹ below 2000 m. These concentrations
decline progressively to approximately 42 µmol kg⁻¹ toward the southern Atlantic, in agreement with the



classical pattern of Atlantic meridional overturning circulation. However, the modeled deep DOC

concentrations in both the Atlantic and Indian Oceans tend to be higher than observed values, potentially

reflecting limitations in the representation of DOC degradation or ventilation processes. In contrast, the

modeled DOC concentrations in the deep Pacific (~40 µmol kg$^{-1}$) are consistent with observations,

reflecting the progressive degradation of DOC during the northward flow of deep waters from the

Southern Ocean and their subsequent return southward through the mesopelagic layer (Hansell, 2013).

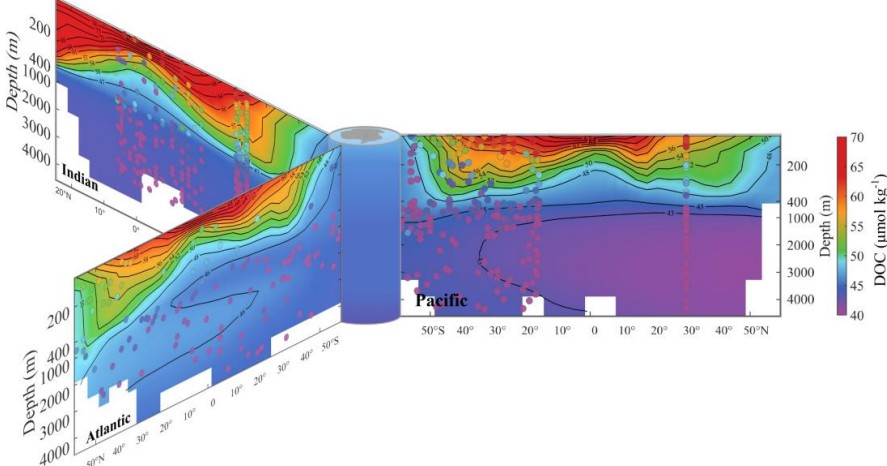


Figure 12. Distributions of DOC (µmol kg$^{-1}$) along meridional sections in the central Atlantic, central
Pacific, and western Indian oceans, respectively. The shaded represents cGENIE-MCP model outputs,
overlaid with scatter points of observed values from Hansell's laboratory (Hansell, 2013).

**4 Discussion**

**4.1 Effect of DOC partitioning**

The distribution of DOC and its constituent fractions provides essential insights into the production,

degradation, and transport processes that govern the oceanic carbon cycle. The cGENIE-MCP model

effectively simulates the vertical transformation of labile DOC (LDOC), which exhibits a global mean

surface concentration of approximately 10 µmol·kg$^{-1}$. The rapid microbial consumption makes LDOC

concentrations approach zero below the euphotic zone (Fig. 13). This rapid cycling underscores LDOC's

role in supporting microbial loop dynamics in surface waters. LDOC is also primarily exits in equatorial

regions, further reflecting its rapid turnover driven by the microbial loop process. SLDOC is abundant in

the euphotic zone, particularly in tropical and subtropical regions, where it supports heterotrophic

bacterial carbon demand. It is laterally redistributed by subtropical gyres and is typically exported to the



upper mesopelagic zone (100-1000 m) due to its moderate lability. With a lifetime of months to years, SLDOC is either remineralized or converted into RDOC during its transit. The simulated spatial extent of SLDOC extends slightly deeper and spans a broader latitudinal range, particularly in the subtropics, suggesting its role as a transitional carbon reservoir that can be exported through physical processes such

as subduction and mesoscale eddy transport.

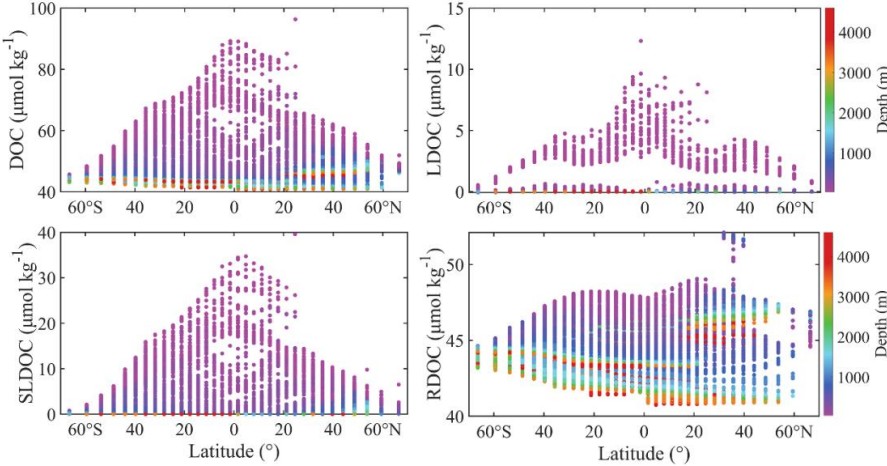

Figure 13. Meridional distributions of mean DOC concentrations (A), mean LDOC concentrations (B), mean SLDOC concentrations (C), mean SLDOC concentrations (D) ($\mu mol\ kg^{-1}$).

In contrast, RDOC is ubiquitous throughout the ocean interior, with relatively uniform concentrations ranging from 40 to 50 $\mu mol \cdot kg^{-1}$ in deep waters. This homogeneity reflects its recalcitrance to microbial degradation and highlights its significance as a long-term carbon reservoir. Slightly elevated RDOC concentrations in high-latitude deep waters may reflect entrainment of surface-derived labile and semi-labile DOC during deep water formation and ventilation. These results underscore the dynamic role

of LDOC and SLDOC as intermediaries in the microbial carbon pump (MCP), mediating the transformation of labile carbon into more persistent forms that contribute to long-term carbon sequestration.

    DOC concentrations are highest in the tropical and subtropical surface waters and decrease with depth and latitude. This pattern reflects enhanced primary production in low-latitude regions and limited

vertical mixing due to strong stratification, allowing DOC to accumulate in surface layers. In contrast, high-latitude regions exhibit lower DOC concentrations due to weaker stratification and lower biological production. The distribution of DOC reflects the functional partitioning of the oceanic carbon reservoir-



from rapidly cycling, bioavailable fractions to long-lived, deeply sequestered refractory pools-
highlighting the essential roles of biological production, microbial transformation, and physical transport
in shaping carbon dynamics and supporting long-term ocean carbon sequestration.

In this study, the cGENIE-MCP model has been extended to incorporate temperature-dependent
nutrient uptake and remineralization of sinking organic matter. These enhancements allow the model to
more accurately capture observed DOC distributions and improve the representation of spatial and
vertical gradients. Furthermore, integrating temperature-sensitive DOC cycling processes enables the
cGENIE-MCP to simulate the dynamic feedbacks between biological activity, ocean carbon sequestration,
and climate warming (Crichton et al., 2021). Partitioning DOC into LDOC, SLDOC, and RDOC provides
a mechanistic framework for evaluating the efficiency of the microbial carbon pump under future climate
scenarios and contributes to a deeper understanding of the ocean's role in long-term carbon storage.

## 4.2 Distribution of DOC production


Primary production (PP) is driven by the availability of nutrients and light within the euphotic zone,
where phytoplankton convert inorganic carbon into organic matters. The spatial distribution
characteristics of PP arise from abiotic controls (e.g., light, nutrient upwelling, stratification) and biotic
feedbacks (e.g., grazing pressure, microbial loop), shaping global productivity gradients from nutrient-
rich coastal upwelling zones to oligotrophic gyres. The DOC originates from biologically mediated
processes in the euphotic zone including phytoplankton extracellular release, top-down pressures (grazers,
bacterial and viral lysis), and bottom-up recycling (particle solubilization, prokaryotic excretion). Spatial-
temporal variations in DOC emerge from the interplay of theses biological and abiotic drivers.

Figure 14 illustrates the spatial distributions of PP and DOC production (LDOC, SLDOC, RDOC)
simulated by the cGENIE-MCP model, revealing region-specific variability driven by nutrient supply,
light availability, hydrodynamic processes, and biological community structure. In equatorial-tropical
ocean regions, PP and LDOC production ($P_{LDOC}$) remain relatively high, reaching values of 400-1000 mg
C m$^{-2}$ d$^{-1}$ and 300-800 mg C m$^{-2}$ d$^{-1}$, respectively (Fig. 14). The spatial coherence between high $P_{LDOC}$ and
high-PP zones indicates that LDOC is primarily derived from PP. SLDOC production ($P_{SLDOC}$) ranges
from 20 to 50 mg C m$^{-2}$ d$^{-1}$, closely mirroring PP and $P_{LDOC}$ patterns, reflecting a progressive attenuation
of DOC bioavailability from labile to semi-labile fractions. RDOC production ($P_{RDOC}$), ranging from 0.4
to 0.8 mg C m$^{-2}$ d$^{-1}$ and constituting approximately 0.1-0.2% of PP, exhibits a relatively uniform spatial
distribution, consistent with Hansell (2013) and Legendre et al. (2015) studies (Table S3). This uniformity



underscores RDOC's role in long-term biogeochemical cycling and carbon sequestration.


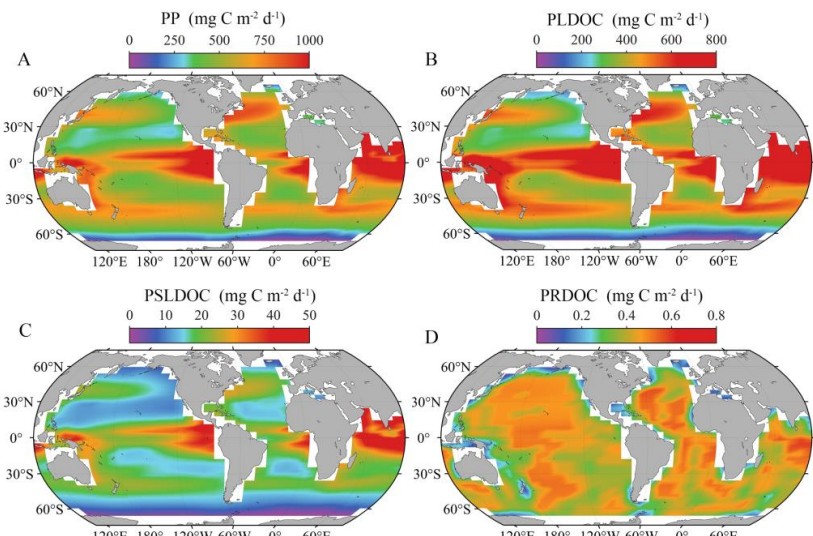

Figure 14. Glogal distributions of the production rate for (A) DOC, (B)LDOC, (C)SLDOC, (D)RDOC (mg C m⁻² d⁻¹), respectively.

## 5 Perspectives


Several "Snowball Earth" events occurred throughout geological history. According to the Snowball Earth hypothesis, the biogeochemical cycle and the PP have severely slowed down or even stagnated under global freezing conditions. However, previous studies have found PP and DOC reservoirs still exist during glaciations (Jiao et al., 2024a; Man et al., 2024). These results suggest that organic matter produced

in the surface ocean may have been degraded into DOC or RDOC in the water column. Therefore, understanding the changes of the DOC pools during Snowball Earth periods is of great significance (Hoffman and Schrag, 2002; Hoffman et al., 2017; Sansjofre et al., 2011). Nevertheless, the storage, spatial distribution, and source-sink processes of DOC during "Snowball Earth" remain largely unexplored. The cGENIE-MCP model proposed in this study can simulate the DOC dynamics over

geological historical spans and explore the relationship between DOC dynamics and long-term climate change. It is possible to analyze the relationship between $\delta^{13}C$ negative excursion and carbon pumps in geological records on a global scale, quantify the efficiency of MCP, and evaluate the impact of ocean environmental changes on the distribution of DOC.



Furthermore, reducing emissions and enhancing carbon sinks have become a global consensus in response to global warming, with ocean carbon sinks playing a vital role in achieving this goal. Previous studies have pointed out that both "Snowball Earth" events and glacial-interglacial cycles are not only driven by orbital forcing but are also influenced by the ocean carbon cycle (Hoffman et al., 2017). The primary reason is due to the huge RDOC, which can remain in the ocean for tens of thousands of years and also release carbon into the atmosphere, thereby regulating climate change. Therefore, evaluating the efficiency of the MCP is crucial for understanding of long-term climate regulation. The cGENIE-MCP model offers selectable modular combination methods and is suitable for long-term climate research. For example, by combining SSP scenarios, the response of MCP to $CO_2$ concentration will be investigated and reveal the feedback mechanism between climate change and the ocean carbon cycle. Evaluating the application potential of ocean negative carbon emission technologies (e.g., ocean alkalinization enhancement technology) under different climate scenarios. Analyzing the long-term effects of ocean negative carbon emission technologies on the environment by simulating different implementation pathways, and comprehensively evaluating the optimal implementation scenarios and strategies for the coupled of ocean carbon pump.

**6 Conclusions**

This paper developed an RDOC parameterized-based method of the MCP process and embedded a temperature-dependent process in cGENIE model and successfully incorporated RDOC as a tracer. The model outputs towards cGENIE-MCP and cGENIE were evaluated and verified the reasonable simulation performance of the models via WOA 2023 products and GLODAP datasets. We derive the spatial distribution of concentrations and production rates of LDOC, SLDOC, and RDOC. The LDOC and SLDOC were hardly accumulated in the surface layer due to their rapid degradation and exhibited a lower concentration than RDOC. The production rates reveal the spatial distribution of LDOC, SLDOC, and RDOC strongly associated with primary production and indicate the critical role of bottom-up control driven by the nutrient cycle in pelagic ecosystems and the primary production process. This study demonstrated that RDOC is the most vital component contributing to carbon sequestration. With this new extension, there is a potential for further study regarding the role of RDOC during future and past climatological perturbations and the role in the biological pump.

**Code availability**



The code of cGENIE-MCP model used in this paper is available at
https://doi.org/10.5281/zenodo.15734206 (Ma and Lai, 2025).

Configurations for the specific experiments presented in the paper can be found in the file
README.md at https://doi.org/10.5281/zenodo.15734206 (Ma and Lai, 2025).

**Author contributions**

WM and NJ designed the MCP module. YL and WM developed the code, ran, and analysed the model
ensemble and data, and all authors co-wrote the paper.

**Acknowledgements**

We are grateful to Prof. Weilei Wang who provided parameters on DOC transformation.

**Financial support**

This study was supported by the Ocean Negative Carbon Emissions (ONCE) Program. National. Key
Research and Development Program of China (2023YFF0805004), National Natural Science Foundation
of China (No. 42276063).

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
