# Peer review of "Incorporating Recalcitrant Dissolved Organic Carbon and Microbial Carbon Pump Processes into the cGENIE Earth System Model (cGENIEv0.9.35-MCP)"

_EGUsphere, 2025_

## Referee Comment (RC1)

Lai et al., present a new model of the Microbial Carbon Pump (MCP) within the Earth System Model cGENIE. The key novel advancement over other models of DOC is linking the production of recalcitrant DOC (RDOC) to the degradation of semi-labile DOC (SLDOC). cGENIE is a good choice for developing a model of the MCP as it is well suited to quantifying the long timescales needed and is widely applied to past climates where the role of RDOC is potentially important.

The manuscript needs substantial improvement in two key areas. Firstly, the model description has a number of potential errors that need addressing to ensure the manuscript accurately describes the model developments. Unfortunately, I couldn't get the code to compile so this is based on reading the code in the short of amount of time for reviewing so my apologies if anything is missed! Secondly, the authors need to demonstrate that their new development is a significant advance on previous models. Models of DOC can equally replicate observed [(R)DOC] in the deep ocean despite resolving radically different processes, e.g., Lennartz et al., (2024) vs. Hansell et al., (2012). The authors need to demonstrate the significance of their new MCP development beyond just matching the modern distribution, i.e., does it behave differently to previous models?

**Potential issues with the model description**

1) **Model description does not seem to match implementation:**

   a. Lines 126 – 127: "*SLDOC degradation is driven primarily by photodegradation*" – this contrasts against the equation on line 215 stating it is only a function of $\tau_{SL}$. The photodegradation option (bg_ctrl_bio_remin_RDOM_photolysis) in the model configuration file provided is commented out and defaults to off (false) in the definition.xml file.

   b. The gas transfer velocity equation does not match that in Ridgwell et al., (2007) as stated. Parameters l, a, b, c and the 0.25 do not appear in that paper or in genie's piston velocity code. This may be referring to combined processes in the model but I cannot find the a, b, c and 0.25 parameters in the code. The value for the scaling factor "l" has a value of 2.778e-6 which seems like the preindustrial molar ratio of $CO_2$ in the atmosphere but I cannot see how this relates to gas transfer velocity specifically.

   c. Line 159: "*Eppley's initial values, with a = 0.59 and b = 75.80*" – the model default value for b (par_bio_kT_eT) is 15.8. These values appear unchanged in the model configuration file or elsewhere in the code, so I can't see how 75.80 is arrived at.

   d. Lines 169 – 173: "*v=0.66*" – the configuration file has selected the Dunne et al., (2005) option for the particle export ratio (bg_opt_bio_red_DOMfrac='dunne') which is an empirical function dependent on temperature, productivity and euphotic zone depth. This looks like it overrides the default global constant value for v of 0.66, instead creating a spatially variable ratio of particle export to total export (see loc_rPOC on line 991 in biogem_biox.f90) from which v (ratio of DOC

production to total: 1-locrPOC) is calculated. The Dunne et al., (2005) scheme is also described in the manuscript.

e. *"While some POC sinks to the bottom waters and is eventually buried in sediments, a substantial fraction is remineralized within the water column."* – this model has no sediment module enabled and therefore has a reflective sediment boundary where all POC hitting the seafloor is remineralised in the overlying grid-box (see Ridgwell et al., 2007).

f. Table 1:
    i. Scaling factor "l" does not obviously appear in Crichton et al., (2021) as described here.
    ii. The half saturation constant values do not seem to match those selected in the configuration file (bg_par_bio_c0_PO4=0.1E-6 and bg_par_bio_c0_Fe=0.10E-09 for PO4 and Fe respectively). I'm not sure of the relevance to the Crichton or Matsumoto references here either.
    iii. v: see comment d above, the configuration file appears to choose a spatially variable scheme not a global fixed value.

g. Figure 11: From the equations for LDOC, SLDOC and RDOC production, these are produced in a globally constant ratio of 0.9599:0.04:0.0001. This is also hard-coded in lines 997 – 999 in biogem_box.f90. Therefore, I would expect the LDOC and SLDOC panels at least to have the same spatial variability but scaled in magnitude. I'm also unsure from the text if the RDOC panel includes production from the SLDOC remineralisation.

h. There is no information on the spin-up protocol. The readme provided suggests the model is spun-up for 30k years from initial conditions. If true, this doesn't seem like sufficient time for RDOC to reach steady-state given its lifetime of 16k years.

2) **Model description of new developments is ambiguous and/or incomplete:**

a. POC remineralisation should have a sinking rate described.
b. SLDOC and RDOC remineralisation rates (k) are described as temperature-dependent yet the equations state that k = 1/tau. 1/tau also seems to be applied in the model code.
c. The transformation of SLDOC to RDOC is unclear as to whether the SLDOC is transformed to a RDOC remineralisation flux or to the RDOC tracer. I think the latter is the case from the code but the naming of flux terms is ambiguous.
d. Ideally there should be differential equations for LDOC, SLDOC and RDOC to fully describe the new DOC cycling in the model
e. The degradation rate constant for resolving temperature-dependent DOC (par_bio_remin_DOC_K1) has been changed from the value in Crichton et al., (2021). This should be included in Table 1.

**Significance of MCP model development**

A key feature of this model is the transformation of SLDOC to RDOC via parameter a, which distinguishes it as a representation of the Microbial Carbon Pump vs. other models of DOC cycling where different pools are unconnected (e.g., MESMO3). The manuscript presents the validation of DOC against observations, but the deep [(R)DOC] can be reproduced by very different models (Lennartz et al., 2024 vs. Hansell et al., 2012). This is because rates of RDOC cycling are smaller than circulation rates leading to a near-uniform distribution, such that the key constraint is the inventory which primarily constrains the magnitude of global fluxes in and out. To demonstrate this is a novel development, what new dynamics does cGENIE-MCP resolve?

One way to demonstrate this would be to show parallel results from the model when a=0. For example, it would be illustrative to see what happens to the DOC inventory across a range of perturbations with a=0.015 and a=0. This could be different atmospheric $CO_2$ values, differences in the DOC production or lifetimes, or differences in total production.

My concern is that the implementation of the MCP isn't significantly different to previous models. To illustrate this, I've used a steady-state analysis (see below). The production of RDOC in other models ($Fin_{RDOC}^{fix}$) is typically a fixed fraction of total DOC production:

$$Fin_{RDOC}^{fix} = \Gamma * v * f_{RDOC} \qquad (1)$$

where $f_{RDOC}$ is the fraction of RDOC production (e.g., equivalent to $1 - f1 - f2$ in this manuscript) and the parameters follow those in this manuscript. For clarity I have treated this as already integrated over the surface layer, e.g., units of Pg C and Pg C yr⁻¹. In the cGENIE-MCP model the production of RDOC has an additional term reflecting the production arising from SLDOC remineralisation:

$$Fin_{RDOC}^{MCP} = \Gamma * v * f3 + a * \frac{SLDOC}{\tau_{SL}} \qquad (2)$$

Assuming steady-state, equation (2) can be rewritten in a comparable form to eqn (1):

$$Fin_{RDOC}^{MCP} = \Gamma * v * (f3 + a * f2) \qquad (3)$$

where $f2$ is the fraction of SLDOC production from total DOC production. If production and the lifetime of RDOC are similar between models then $f_{RDOC} \equiv (f3 + a * f2)$ to achieve the modern inventory of RDOC at steady-state. In cGENIE-MCP the parameters $f2, f3, a$ and $v$ are fixed values (note the question about v above however). As such, the two different models (equations (1) and (3)) can be seen to have similar steady-state dynamics where the dynamic term is export production ($\Gamma$) which is scaled by a fixed fraction. The steady-state constraint (eqn 6) also implies that RDOC may not be sensitive to a change in $\tau_{SLDOC}$.

This is a simple analysis which doesn't account for different nutrient feedbacks between SLDOC and RDOC cycling but hopefully it encourages the authors to demonstrate that their model *behaves* in a novel way to other models. What happens if you have a change

in SLDOC production or remineralisation? Do the models behave differently for the same change in atmospheric $CO_2$? If this has a notable effect on RDOC, which would not be the case in previous models, then this model would be a significant advancement in resolving DOC dynamics.

**Choice of results to evaluate the model**

Overall, the results section is heavily weighted towards assessing major ocean tracers not DOC. I understand the authors want to demonstrate their new development doesn't degrade overall model performance but arguably adding a component of organic matter cycling that is characterised by relatively small rates is probably not likely to perturb tracers like PO4, $O_2$ or DIC that much. Table 2 in the manuscript summarises this immediately such that the following text and figures don't add much more highly relevant information. I would suggest minimising this part of the results substantially and/or relocating to supplementary.

It is useful to show temperature as it determines some rates in the model but the only feedback between ocean biogeochemistry and the physical model I'm aware of in cGENIE is atmospheric $CO_2$, which here is restored to a preindustrial value in these runs. Therefore, temperature does not need evaluating to the extent it is here.

The description should increase its evaluation of DOC in the model since this is the key focus of the new development. The existing evaluation is much briefer and less quantitative for DOC than of other tracers limiting comparison against previous developments and current constraints. Some of this appears in supplementary and would be more informative in the main text. It would be good to see some standard metrics comparable to Table 1 in Hansell (2013) such as inventory (Pg C) production rate (Pg C $yr^{-1}$), removal rate, lifetime (years) for each of the DOC components. It would also be helpful to see the cGENIE and cGENIE-MCP comparison which is shown for other tracers but not DOC – is the labile DOC similar between models? It's notable that isotopes are omitted in this manuscript when one of the main advantages of cGENIE is its ability to resolve isotopes! The bulk radiocarbon age is a key constraint on DOC cycling which could be added to the evaluation.

**Functionality of the model**

Several things make the model less functional or harder to use from the description:

a) The tracer names are misaligned between the manuscript and model (LDOC = DOC; SLDOC = RDOC, RDOC = URDOC in the manuscript and model respectively). Ideally, these should be the same.
b) Some MCP parameters are hard-coded (f1, f2, a) which reduces the ability of users to explore the MCP. In your discussion you discuss being able to explore the dynamics of the MCP in detail with this model but not being able to change these parameters is a very big limitation to this.

c) The activation energy of LDOC is hard-coded to the labile POC parameter so there is no ability to decouple these. Is it reasonable to assume POC and LDOC will be always treated the same?

**Evaluation and presentation**

a) The projection choice for Figure 2 limits comparison against other figures. It has no longitude or latitude values.
b) The RMSE presented is actually the centered-RMSE which is the underlying statistic for the Taylor diagram (Jolliff et al., 2009). It would be more informative to show a Taylor diagram so we can assess the spatial distribution and variability of tracers between the two model versions.
c) Please also consider other misfit functions to assess the model. Kriest et al., (2010) provides a good overview of approaches as well as the impact of volume-weighting. This discussion may be relevant here because DOC is considerably more variable in the upper ocean than the deep ocean which these alternative functions can deal with.
d) Why are there no quantitative analysis of the DOC comparison, e.g., RMSE?
e) What is the reason for using specific regions for the model-data comparison? These have been changed from those used in Crichton et al., (2021) with no justification beyond "limited relevance or observational data availability" (lines 251-252). What does this mean and what is the basis for choosing these regions?
f) What do the errorbars represent in the vertical profiles?
g) The model description does not clearly distinguish between *new* developments to cGENIE and previous developments. I would suggest to minimise descriptions of existing processes such as air-sea gas exchange, unless they provide important context, to avoid assumptions that they are included in the new developments of cGENIE.
h) There are numerous descriptions of the model fit to observations throughout that aren't supported quantitatively, e.g., "good accuracy" "show good agreement" "more accurately" "moderate discrepancies" "approximately reasonable". This language should be modified unless it is directly related to a quantitative measure.

**Specific Comments**

Line 56: "observed RDOC" – more accurately this should be observed deep-ocean bulk [DOC] as there are competing hypotheses about whether this is labile or a mixture of compounds with different reactivities, e.g., Follett et al., (2014).

Lines 59 – 60: The deep ocean [DOC] and radiocarbon signature issues can be alleviated by adding a simple RDOC pool without MCP parameterisations. The MCP parameterisations add specific dynamics related to why the RDOC accumulates.

Lines 78 – 81: It would be good to explicitly explain how MESMO represents SLDOC and RDOC to enable direct comparison with cGENIE-MCP here.

Lines 103 – 104: The physical circulation parameters are derived from Cao et al., (2009) configuration but you are using a different continental grid to them (worlg4 vs worjh2). I think the Ward et al., (2018) citation might be more appropriate.

Lines 119 – 120: Though some details in that paper are relevant here, Ward et al., (2018) describes the trait-based ecosystem so it might help to clarify this distinction here. Are the parameter values or equations used developed in the Matsumoto or Tanioka papers?

Line 121: "particulate organic matters (POMs)" – usually this would just be singular not plural.

Line 163: "primary production" – GENIE resolves net export production which is equivalent to net community production (NCP) across large enough spatial scales and temporal scales. Primary production is ambiguous to this difference and should probably be avoided.

Line 228: "a is the conversion rate" – this does not have units of a rate

Lines 310 -311: or equally that the addition of RDOC has a negligible effect on the large-scale $PO_4$ distribution?

Lines 347 – 350: A more appropriate experiment would be a run forced with historical atmospheric $CO_2$ concentrations from the preindustrial to present-day continuing from your preindustrial spin-up. Or alternatively compare against the GLODAP DIC observations with the anthropogenic DIC component removed (Cant in GLODAP).

Line 371: "DOC is a key component of ocean carbon cycling" – maybe my comment is not exactly relevant to here but this refers to multiple things. The recycling of labile DOC is crucial for models to resolve productivity. RDOC is potentially important for carbon storage though this is subject to the timescale discussed.

Lines 377 – 380: "cGENIE-MCP model exhibits improved agreement with observed DOC distributions in both surface and deep layers [compared to MESMO 3]" – please quantify this! Can you get the MESMO-3 results and compare concentrations, distributions, RMSE? Otherwise, this is an unqualified statement that cannot be verified.

Lines 398 – 403: The supplementary figures and tables (Fig S6, Tables S2 and S3) need to be in the main text as these are essential comparisons against observations and models of DOC, which is the key focus of this manuscript.

Line 441: "lifetime of months to years" – you have prescribed a fixed lifetime in the model, it is not variable.

Figure 13: This is an unusual way of plotting meridional distributions which is pretty hard to interpret. It would be much clearer to show zonal averages for each DOC pool such as in Figure 12.

Lines 452 – 454: "Slight elevated RDOC concentrations ... may reflect entrainment of surface-derived labile semi-labile DOC..." – is it possible to back this out of the model and demonstrate?

Lines 454 – 457: "These results underscore the dynamic role of LDOC and SLDOC" – I don't think this statement is supported because you have only shown a steady-state results. To support this you need to show that the model behaves differently to some perturbations compared to the model without the interactions between DOC pools that underpin the MCP concept.

Section 4.2: This version of GENIE resolves net export production not primary production. It has no representation of some of the processes described here like top-down grazing pressure, bacterial and viral lysis, particle solubilization – the net effect of these processes are parameterised by the Michaelis-Menten uptake scheme you are using. This discussion should be amended to better reflect what the model is actually doing and how it represents the complex reality.

Line 518: "huge RDOC" – please quantify this! The DOC inventory is around 700 Pg C whereas the regenerated DIC pool from the Biological Carbon Pump is around 1700 Pg C which undermines this argument. I would argue it is the potential dynamics that are crucial here – how likely, and by how much, could RDOC and regenerated DIC change in response to perturbations or different factors?

Line 529: "coupled of ocean carbon pump" – it's not clear what this is referring to.

Figure S6: Why is the Sargasso sea singled out here and is this modelled or observed concentrations? What area does panel B correspond to – global average? Can the comparison against Wang et al., (2023) be expanded?

Tables S2 and S3: This is useful but seems like it could easily expanded. Could you add comparison against MESMO here? Are there other datasets and models that could be compared against? Can the analysis be expanded to more regions?

**References**

Follett et al., (2014) Hidden cycle of dissolved organic carbon in the deep ocean *Proceedings of the National Academy of Sciences*, Vol. 111, No. 47 p. 16706-16711

Hansell et al., (2012) Net removal of major marine dissolved organic carbon fractions in the subsurface ocean. *Global Biogeochemical Cycles*, Vol. 26, No. 1

Jolliff et al., (2009) Summary diagrams for coupled hydrodynamic-ecosystem model skill assessment. *Journal of Marine Systems*, Vol. 76, No. 1 – 2, p. 64 – 82

Kriest et al., (2010) Towards an assessment of simple global marine biogeochemical models of different complexity. *Progress in Oceanography*, Vol. 86, No. 3—4, p. 337 – 360

Lennartz, S. T., Keller, D. P., Oschlies, A., Blasius, B., & Dittmar, T. (2024). Mechanisms underpinning the net removal rates of dissolved organic carbon in the global ocean. Global Biogeochemical Cycles, 38, e2023GB007912. https://doi.org/10.1029/2023GB007912

**Steady-state analysis**

Using the same parameters in the manuscript but expanding the remineralisation terms and assuming this is integrated across the surface layer for clarity, the equation for the production of RDOC is:

$$F_{z=he}^{RDOC} = \Gamma * v * f3 + a * \frac{SLDOC}{\tau_{SLDOC}} \tag{4}$$

We can assume steady-state for the governing SLDOC differential equation ($\frac{dSLDOC}{dt} = 0$) and rearrange to find an expression for $\frac{SLDOC}{\tau_{SLDOC}}$ (eqn 6):

$$\frac{dSLDOC}{dt} = \Gamma * v * f2 - \frac{SLDOC}{\tau_{SLDOC}} \tag{5}$$

$$\frac{SLDOC}{\tau sdoc} = \Gamma * v * f2 \tag{6}$$

Eqn (6) can be substituted into eqn (4) and simplified to get an expression for RDOC production that is comparable to previous models:

$$F_{z=he}^{RDOC} = \Gamma * v * (f3 + a * f2) \tag{7}$$

---

## Author Comment (AC2)

**Response to reviewer comments**

We would like to thank the reviewer for their thorough evaluation of our manuscript and for the constructive comments and suggestions. We have carefully revised the manuscript according to the comments. In the following, we provide a detailed, point-by-point response to all the comments. For clarity, the comments are presented in italics, followed by our responses. All changes made in the manuscript are highlighted in the revised version.

**Potential issues with the model description**

- 1) Model description does not seem to match implementation:
- a. Lines 126 127: "SLDOC degradation is driven primarily by photodegradation" this contrasts against the equation on line 215 stating it is only a function of  $\tau$ !". The photodegradation option (bg\_ctrl\_bio\_remin\_RDOM\_photolysis) in the model configuration file provided is commented out and defaults to off (false) in the definition.xml file.

**Response:**

After carefully re-checking the model configuration files, we realized that original description was inaccurate. In our simulations, the photodegradation option (bg\_ctrl\_bio\_remin\_RDOM\_photolysis) is indeed commented out in the configuration file and defaults to "false" in the definition.xml. Therefore, SLDOC degradation is not driven by photodegradation in the current model setup. Instead, it is represented by a prescribed lifetime  $(\tau)$ , as shown in the equation on Line 210. Accordingly, we have revised the sentence in **Lines 129-130** to: "LDOC degradation is temperature dependent, SLDOC degradation is represented by a prescribed lifetime  $(\tau)$ , and a fraction of SLDOC is converted into RDOC."

b. The gas transfer velocity equation does not match that in Ridgwell et al., (2007) as stated. Parameters l, a, b, c and the 0.25 do not appear in that paper or in genie's piston velocity code. This may be referring to combined processes in the model but I cannot find the a, b, c and 0.25 parameters in the code. The value for the scaling factor "I" has a value of 2.778e-6 which seems like the preindustrial molar ratio of CO2 in the atmosphere but I cannot see how this relates to gas transfer velocity specifically.

**Response:**

The parameters l, a, b, c, and 0.25 do not appear in Ridgwell et al. (2007). These terms were originally taken from Ridgwell (2001), and we mistakenly attributed them to the latter. To avoid confusion, we have revised the manuscript to present the standard formulation of the air-sea CO2 gas transfer velocity following Wanninkhof (1992), which is consistent with the cGENIE model implementation. The corrected description is now given in the Supporting Information (Section 1). In the revised text, we now clearly state that: 'CO2 solubility and the Schmidt number are parameterized following

Wanninkhof (1992); The gas transfer velocity k follows the standard quadratic windspeed dependence with a scaling factor of 0.31.' To improve clarity and avoid over complicating the main text, the detailed formulation and parameter definitions have now been moved to the Supporting Information. The unnecessary parameters that caused confusion have been removed.

c. Line 159: "Eppley's initial values, with a = 0.59 and b = 75.80"-the model default value for b (par\_bio\_kT\_eT) is 15.8. These values appear unchanged in the model configuration file or elsewhere in the code, so I can't see how 75.80 is arrived at.

**Response:**

In the cGENIE configuration and code, the default value is b = 15.8 (parameter par\_bio\_kT\_eT), which is the value actually used in our simulations. We have corrected the manuscript accordingly (**Line 145**), and now state: "The temperature dependence of biological processes follows Eppley's formulation, with a = 0.59 and b = 15.8." This correction does not affect the simulations or results presented in the paper.

d. Lines 169-173: "v=0.66"-the configuration file has selected the Dunne et al., (2005) option for the particle export ratio (bg\_opt\_bio\_red\_DOMfrac='dunne') which is an empirical function dependent on temperature, productivity and euphotic zone depth. This looks like it overrides the default global constant value for v of 0.66, instead creating a spatially variable ratio of particle export to total export (see loc\_rPOC on line 991 in biogem\_box.f90) from which v (ratio of DOC production to total: 1-locrPOC) is calculated. The Dunne et al., (2005) scheme is also described in the manuscript.

**Response:**

The original text stated a fixed value of v=0.66, corresponding to the default OCMIP-2 setting in cGENIE (Ridgwell et al., 2007). In our experiments, we employed the Dunne et al. (2005) scheme (bg\_opt\_bio\_red\_DOMfrac='dunne'), in which v is dynamically calculated as v = 1 - rPOC, rPOC is an empirical function of temperature (T), where rPOC = 0.419 + 0.0582 ln(PP/Zeu)-0.0101T, bounded between 0.04 and 0.72. We have revised the manuscript (**Lines 161-163**) to clarify this, and the corrected description now explicitly states that v is spatially variable rather than constant. This modification ensures consistency between the model description and the actual implementation.

e. "While some POC sinks to the bottom waters and is eventually buried in sediments, a substantial fraction is remineralized within the water column." – this model has no sediment module enabled and therefore has a reflective sediment boundary where all POC hitting the seafloor is remineralised in the overlying grid-box (see Ridgwell et

**Response:**

We thank the reviewer for pointing out this important clarification. The original sentence was misleading because the sediment module was not enabled in our model configuration. In our simulations, POC reaching the seafloor is indeed remineralized in the bottom water grid box, following the reflective sediment boundary condition of cGENIE (Ridgwell et al., 2007), and not buried in sediments. Accordingly, we have revised the text to read: "POC sinks to deep water and is remineralized within the water column, returing carbon to the overlying bottom water grid box." (Lines 191-192)

**f. Table 1:**

- i. Scaling factor "l" does not obviously appear in Crichton et al., (2021) as described here.
- ii. The half saturation constant values do not seem to match those selected in the configuration file (bg\_par\_bio\_c0\_PO4=0.1E-6 and bg\_par\_bio\_c0\_Fe=0.10E-09 for  $PO_4$  and Fe respectively). I'm not sure of the relevance to the Crichton or Matsumoto references here either.
- iii. v: see comment d above, the configuration file appears to choose a spatially variable scheme not a global fixed value.

**Response:**

We revised **Table 1** and removed the entries for the scaling factor "l" and the DOC fraction "v" as suggested in earlier comments. We also acknowledge that in the originally submitted manuscript, the half-saturation constants for PO4 and Fe listed in Table 1 did not match those used in the model configuration file. In the revised manuscript, the half-saturation constants now match the configuration file values (bg\_par\_bio\_c0\_PO4 = 0.1E-6 and bg\_par\_bio\_c0\_Fe = 0.10E-09). These parameter values are directly inherited from the default cGENIE setup.

g. Figure 11: From the equations for LDOC, SLDOC and RDOC production, these are produced in a globally constant ratio of 0.9599:0.04:0.0001. This is also hard-coded in lines 997–999 in biogem\_box.f90. Therefore, I would expect the LDOC and SLDOC panels at least to have the same spatial variability but scaled in magnitude. I'm also unsure from the text if the RDOC panel includes production from the SLDOC remineralization.

**Response:**

The initial partitioning of DOC production among LDOC, SLDOC, and RDOC is set by fixed fractions (0.9599:0.04:0.0001) in the source code (in biogem box.f90).

Consequently, their surface production patterns are indeed spatially similar. However, the subsequent spatial distributions of LDOC, SLDOC, and RDOC diverge substantially due to their distinct remineralization rates and transformation pathways. Specifically, LDOC is highly labile and rapidly remineralized within the surface and upper thermocline, leading to strong surface gradients and minimal penetration depth. In contrast, SLDOC has a longer lifetime, allowing it to be advected and mixed to intermediate and deep waters before remineralization. As a result, SLDOC exhibits a broader and smoother spatial pattern. For RDOC, the field shown in Fig. 5 includes both the small fraction directly produced at the surface (0.01% of total DOC production) and the additional RDOC generated from SLDOC remineralization through the MCP transformation term (a×[SLDOC]). This source introduces a distributed production pathway throughout the water column, particularly in regions of active remineralization, and is responsible for the gradual accumulation and redistribution of RDOC by ocean circulation. Because of its very long lifetime (16000 years), RDOC becomes nearly homogeneous at the global scale, consistent with its weak spatial variability in the figure. Thus, while surface production ratios are fixed, the emergent global DOC distributions reflect the combined effects of distinct decay constants, transformation coupling (parameter a), and advective-diffusive transport, rather than simple proportional scaling of the initial production fields.

h. There is no information on the spin-up protocol. The readme provided suggests the model is spun-up for 30k years from initial conditions. If true, this doesn't seem like sufficient time for RDOC to reach steady-state given its lifetime of 16k years.

**Response:**

We restarted the model for further 70k years and plot the curve of global mean RDOC concentration with time (the figure below). RDOC concentration almost reach the steady state after the 60,000th year. Therefore, we added the information of mode spin-up in **Lines 114-115**: "The model was run for 100.000 years to reach a steady state, and the result of the last year was used for analysis." The README file has been updated accordingly.

**Surface RDOC 52.20 51.20 50.20 649.20 547.20 5447.20 5447.20 5447.20 644.20 642.20 6544.20 6542.20 6542.20 6542.20 6542.20 6542.20 6542.20 6542.20 6542.20 6542.20 6542.20 6542.20 6542.20 6542.20 6542.20 6542.20 6542.20 6542.20 6542.20 6542.20 6542.20 6542.20 6542.20 6542.20 6542.20 6542.20 6542.20 6542.20 6542.20 6542.20 6542.20 6542.20 6542.20 6542.20 6542.20 6542.20 6542.20 6542.20 6542.20 6542.20 6542.20 6542.20 6542.20 6542.20 6542.20 6542.20 6542.20 6542.20 6542.20 6542.20 6542.20 6542.20 6542.20 6542.20 6542.20 6542.20 6542.20 6542.20 6542.20 6542.20 6542.20 6542.20 6542.20 6542.20 6542.20 6542.20 6542.20 6542.20 6542.20 6542.20 6542.20 6542.20 6542.20 6542.20 6542.20 6542.20 6542.20 6542.20 6542.20 6542.20 6542.20 6542.20 6542.20 6542.20 6542.20 6542.20 6542.20 6542.20 6542.20 6542.20 6542.20 6542.20 6542.20 6542.20 6542.20 6542.20 6542.20 6542.20 6542.20 6542.20 6542.20 6542.20 6542.20 6542.20 6542.20 6542.20 6542.20 6542.20 6542.20 6542.20 6542.20 6542.20 6542.20 6542.20 6542.20 6542.20 6542.20 6542.20 6542.20 6542.20 6542.20 6542.20 6542.20 6542.20 6542.20 6542.20 6542.20 6542.20 6542.20 6542.20 6542.20 6542.20 6542.20 6542.20 6542.20 6542.20 6542.20 6542.20 6542.20 6542.20 6542.20 6542.20 6542.20 6542.20 6542.20 6542.20 6542.20 6542.20 6542.20 6542.20 6542.20 6542.20 6542.20 6542.20 6542.20 6542.20 6542.20 6542.20 6542.20 6542.20 6542.20 6542.20 6542.20 6542.20 6542.20 6542.20 6542.20 6542.20 6542.20 6542.20 6542.20 6542.20 6542.20 6542.20 6542.20 6542.20 6542.20 6542.20 6542.20 6542.20 6542.20 6542.20 6542.20 6542.20 6542.20 6542.20 6542.20 6542.20 6542.20 6542.20 6542.20 6542.20 6542.20 6542.20 6542.20 6542.20 6542.20 6542.20 6542.20 6542.20 6542.20 6542.20 6542.20 6542.20 6542.20 6542.20 6542.20 6542.20 6542.20 6542.20 6542.20 6542.20 6542.20 6542.20 6542.20 6542.20 6542.20 6542.20 6542.20 6542.20 6542.20 6542.20 6542.20 6542.20 6542.20 6542.20 6542.20 6542.20 6542.20 6542.20 6542.20 6542.20 6542.20 6542.20 6542.20 6542.20 6542.20 6542.20 6542.20 6542.20 6542.20 6542.20 6542.20 6542.20 6542.20 65**

The global mean RDOC concentration with model time.

**2) Model description of new developments is ambiguous and/or incomplete: a. POC remineralization should have a sinking rate described.**

**Response:**

In the cGENIE-MCP configuration used in this study, POC remineralization is associated with an explicit sinking velocity that controls the vertical transport and depth-dependent decay of particulate organic carbon. We have now clarified this in **Section 2.2.3**. The revised text now reads (**Lines192-194**): "POC is transported vertically with a prescribed sinking velocity of 125 m day-1 and is remineralized following an exponential attenuation with depth, governed by the characteristic remineralization length scales (POC1 = 589.9 m, POC2 =  $1 \times 10^6$  m)." These parameters have been included in Table 1 of the revised version for transparency.

**b. SLDOC and RDOC remineralization rates (k) are described as temperature-dependent yet the equations state that $k = 1/\tau$ . $1/\tau$ also seems to be applied in the model code.**

**Response:**

We have clarified in Line 209. In the present simulations, SLDOC and RDOC remineralization rates are implemented as  $k=1/\tau$ , with  $\tau$  being a globally constant value. Although the code provides an option for temperature-dependent remineralization, this feature was not activated in our configuration. The text has been updated to accurately reflect the model implementation. Future work could explore the impact of implementing temperature-dependent remineralization rates on DOC cycling and distribution.

c. The transformation of SLDOC to RDOC is unclear as to whether the SLDOC is transformed to a RDOC remineralization flux or to the RDOC tracer. I think the latter is the case from the code but the naming of flux terms is ambiguous.

**Response:**

Indeed, the transformation from SLDOC to RDOC in our model refers to the conversion between tracers, rather than to a remineralization flux. Specifically, during the organic matter cycling, a fraction of POM degradation is allocated to different DOC tracers (LDOC, SLDOC, and RDOC) through the variables loc\_bio\_red\_DOMfrac, loc\_bio\_red\_RDOMfrac, and loc\_bio\_red\_URDOMfrac. In the subsequent remineralization process, the SLDOC pool can be partly remineralized to inorganic constituents or further transferred into the RDOC tracer. Therefore, the naming of "flux" in the code refers to the mass transfer between tracers, not to remineralization fluxes in the strict sense. To clarify this mechanism, we have revised the text as: "a is a dimensionless conversion coefficient that represents the transformation of SLDOC into RDOC, following the parameterization of Wang et al. (2023). The resulting RDOC tracer undergoes slow remineralization independently." (Lines 220-222).

d. Ideally there should be differential equations for LDOC, SLDOC and RDOC to fully describe the new DOC cycling in the model.

**Response:**

In our model, the cycling of DOC pools (LDOC, SLDOC, and RDOC) is already represented through production, remineralization, and transformation fluxes. To clarify the dynamic relationships among these components, we have added the corresponding differential equations in **Section 2.2.2** to explicitly show how DOC concentrations evolve over time:

$$\begin{split} \frac{\partial [LDOC]}{\partial t} &= F_{prod} \cdot f_1 - k_{zLDOC}[LDOC] \\ \frac{\partial [SLDOC]}{\partial t} &= F_{prod} \cdot f_2 - k_{zSLDOC}[SLDOC] - a \left[SLDOC\right] \\ \frac{\partial [RDOC]}{\partial t} &= F_{prod} \cdot (1 - f_1 - f_2) + a \left[SLDOC\right] - k_{zRDOC}[RDOC] \end{split}$$

where  $F_{prod}$  is the total net export production of organic carbon.  $f_1$ =0.9599 and  $f_2$ =0.04 are the export partitioning coefficients for LDOP and SLDOP, respectively, based on Wang et al. (2023),  $k_{zLDOC}$  is the temperature-dependent remineralization rate of LDOC at z layer;  $k_{zSLDOC}$  is the temperature-dependent remineralization rate of SLDOC at z layer (yr-1);  $k_{zRDOC}$  is the remineralization rate of RDOC at z layer (yr-1); a is dimensionless conversion coefficient that converts SLDOC into RDOC, which is based on the Wang et al. (2023); SLDOC is converted to RDOC tracer, which then undergoes slow remineralization independently. (Lines 224-229)

e. The degradation rate constant for resolving temperature-dependent DOC (par\_bio\_remin\_DOC\_K1) has been changed from the value in Crichton et al., (2021). This should be included in Table 1.

**Response:**

We have updated Table 1 to include the modified value of par\_bio\_remin\_DOC\_K1, with a note indicating that it is adjusted from Crichton et al. (2021) to reflect the temperature-dependent DOC remineralization in our model.

**Significance of MCP model development**

A key feature of this model is the transformation of SLDOC to RDOC via parameter a, which distinguishes it as a representation of the Microbial Carbon Pump vs. other models of DOC cycling where different pools are unconnected (e.g., MESMO3). The manuscript presents the validation of DOC against observations, but the deep [(R)DOC] can be reproduced by very different models (Lennartz et al., 2024 vs. Hansell et al., 2012). This is because rates of RDOC cycling are smaller than circulation rates leading to a near-uniform distribution, such that the key constraint is the inventory which primarily constrains the magnitude of global fluxes in and out. To demonstrate this is a novel development, what new dynamics does cGENIE-MCP resolve?

One way to demonstrate this would be to show parallel results from the model when a=0. For example, it would be illustrative to see what happens to the DOC inventory across a range of perturbations with a=0.015 and a=0. This could be different atmospheric  $CO_2$  values, differences in the DOC production or lifetimes, or differences in total production.

My concern is that the implementation of the MCP isn't significantly different to previous models. To illustrate this, I've used a steady-state analysis (see below). The production of RDOC in other models ( $Fin_{RDOC}^{fix}$ ) is typically a fixed fraction of total DOC production:

$$Fin_{RDOC}^{fix} = \Gamma * v * f_{RDOC} \tag{1}$$

where  $f_{RDOC}$  is the fraction of RDOC production (e.g., equivalent to 1-f1-f2 in this manuscript) and the parameters follow those in this manuscript. For clarity I have treated this as already integrated over the surface layer, e.g., units of Pg C and Pg C yr1. In the cGENIE-MCP model the production of RDOC has an additional term reflecting the production arising from SLDOC remineralization:

$$Fin_{RDOC}^{MCP} = \Gamma * v * f_3 + a * \frac{SLDOC}{\tau_{SL}}$$
 (2)

Assuming steady-state, equation (2) can be rewritten in a comparable form to eqn (1):

$$Fin^{MCP}_{RDOC} = \Gamma * \upsilon * (f_3 + a * f_2)$$

where  $f_2$  is the fraction of SLDOC production from total DOC production. If production and the lifetime of RDOC are similar between models then  $f_{RDOC} = (f3 + a * f_2)$  to achieve the modern inventory of RDOC at steady-state. In cGENIE-MCP the parameters  $f_2$ ,  $f_3$ , a and v are fixed values (note the question about v above however). As such, the two different models (equations (1) and (3)) can be seen to have similar steady-state dynamics where the dynamic term is export production ( $\Gamma$ ) which is scaled by a fixed fraction. The steady-state constraint (eqn 6) also implies that RDOC may not be sensitive to a change in  $\tau_{SLDOC}$ .

This is a simple analysis which doesn't account for different nutrient feedbacks between SLDOC and RDOC cycling but hopefully it encourages the authors to demonstrate that their model behaves in a novel way to other models. What happens if you have a change in SLDOC production or remineralization? Do the models behave differently for the same change in atmospheric  $CO_2$ ? If this has a notable effect on RDOC, which would not be the case in previous models, then this model would be a significant advancement in resolving DOC dynamics.

**Response:**

We agree that under strict steady-state assumptions, the expression  $Fin_{RDOC}^{MCP} = \Gamma * v * (f_3 + a * f_2)$  appears algebraically similar to a fixed-fraction parameterization. However, as the reviewer also notes, the key distinction of the MCP formulation lies in its coupling between DOC pools, rather than in a steady-state balance.

The cGENIE-MCP model introduces a coupling between SLDOC and RDOC through the parameter a, which fundamentally alters the accumulation behavior of the DOC system.

In previous models (e.g., MESMO3: DOC is only divided into two types, namely DOC and RDOC.), DOC and RDOC pools are independent, and the deep DOC inventory depends solely on surface production and a fixed remineralization rate. In contrast, the MCP formulation allows part of the SLDOC to be transformed into RDOC continuously throughout the water column. This introduces a feedback between middepth remineralization and the accumulation of recalcitrant carbon, representing a mechanistic analogue to the process of the Microbial Carbon Pump.

To demonstrate this, we have conducted a set of sensitivity experiments comparing simulations with a=0 and a=0.015 under identical physical and biogeochemical forcings. The results show that:

When a=0, the RDOC concentration becomes nearly vertically uniform and its magnitude decreases substantially (see figures below). When a=0.015, by contrast, the model reproduces a more realistic vertical gradient of RDOC.

The concentration pattern reflects the underlying process: when a=0, RDOC behaves as a passive tracer controlled solely by fraction; when a>0, the model resolves an additional slow carbon transfer pathway from SLDOC remineralization, introducing a delayed but continuous source of RDOC. This feedback alters the residence time and vertical redistribution of DOC, generating emergent behavior that cannot be captured by previous unconnected-pool models (e.g., MESMO3).

This mechanism is conceptually analogous to the advective-diffusive DOC export identified in recent inverse modeling studies (Wang et al., 2023), where semi-labile DOC contributes to long-term carbon sequestration through slow subduction and remineralization. Therefore, the MCP implementation in cGENIE-MCP represents a mechanistic advance: it links biological production, microbial transformation, and physical transport in a unified framework, allowing the model to simulate long-term DOC dynamics and vertical structures that were not represented in earlier steady-state DOC models.

Surface RDOC concentration ( $\mu$ mol kg-1). (A) a = 0.015, (B) when a = 0.

DOC concentration ( $\mu$ mol kg-1) along meridional sections in the central Pacific. (A) a = 0.015, (B) a = 0.

Vertical profiles of RDOC concentration comparing with the results of MESMO3 and Wang et al. (2023). cGENIE-MCP is the result by setting a=0.015, and cGENIE-MCPa0 represents a=0.

**Choice of results to evaluate the model**

Overall, the results section is heavily weighted towards assessing major ocean tracers not DOC. I understand the authors want to demonstrate their new development doesn't degrade overall model performance but arguably adding a component of organic matter cycling that is characterised by relatively small rates is probably not likely to perturb tracers like  $PO_4$ ,  $O_2$  or DIC that much. Table 2 in the manuscript summarises this immediately such that the following text and figures don't add much more highly relevant information. I would suggest minimising this part of the results substantially and/or relocating to supplementary.

It is useful to show temperature as it determines some rates in the model but the only feedback between ocean biogeochemistry and the physical model I'm aware of in cGENIE is atmospheric CO2, which here is restored to a preindustrial value in these runs. Therefore, temperature does not need evaluating to the extent it is here.

The description should increase its evaluation of DOC in the model since this is the key focus of the new development. The existing evaluation is much briefer and less quantitative for DOC than of other tracers limiting comparison against previous developments and current constraints. Some of this appears in supplementary and would be more informative in the main text. It would be good to see some standard metrics comparable to Table 1 in Hansell (2013) such as inventory (Pg C) production rate (Pg C yr1), removal rate, lifetime (years) for each of the DOC components. It would also be helpful to see the cGENIE and cGENIE-MCP comparison which is shown

for other tracers but not DOC – is the labile DOC similar between models? It's notable that isotopes are omitted in this manuscript when one of the main advantages of cGENIE is its ability to resolve isotopes! The bulk radiocarbon age is a key constraint on DOC cycling which could be added to the evaluation.

**Response:**

We have carefully revised the Results section to address these concerns as follows:

1. Rebalancing the focus of the Results section:

We agree that the evaluation of PO4, O2, and DIC in the original version was too extensive relative to the DOC assessment. In the revised manuscript, these analyses have been substantially shortened and relocated to the Supplementary Information (Figures S1-S7). The main text now focuses on DOC and its components, highlighting the model's new developments and their implications for the global carbon cycle.

**2. Temperature evaluation:**

We acknowledge that temperature feedback in the cGENIE framework is primarily via atmospheric CO2, which is restored to preindustrial levels in our experiments. Accordingly, we have reduced the discussion of temperature in the main text and now only include it briefly to illustrate its role in controlling key biogeochemical rate processes. In the revised manuscript, the relevant section now reads as (Lines 271-289):

"Ocean surface temperature plays a crucial role in biogeochemical cycling and nutrient uptake (Yan et al., 2024). To assess the performance of the cGENIE-MCP model, simulated temperature, PO4, O2, and DIC fields were evaluated against observations from the WOA23 and GLODAP at three representative depth levels: surface, intermediate (400 m), and deep (3000 m) layers. As shown in Fig. 3, the cGENIE-MCP model reproduces the large-scale spatial distribution of ocean temperature, capturing major patterns in the surface and low-latitude regions. Temperature decreases from the tropics to the poles and shows little variability in the deep ocean (Fig. 4). The model agrees well with observations, with regional RMSEs of 0.6-1.2 °C (Table 2). Minor biases occur in high-latitude, deep, and upwelling regions, likely due to simplifications in the atmospheric component and limited upper-ocean stratification.

The cGENIE-MCP model also effectively reproduces the large-scale spatial and vertical distributions of PO4, O2, DIC, and alkalinity compared with WOA23 and GLODAP datasets (Table 2 and Fig. S1-S7). Modeled PO4 concentrations (0-4.5 μmol kg-1) and vertical profiles closely match observations, with an overall RMSE of ~0.1 μmol kg-1 (Fig. S1-S2). Simulated dissolved oxygen (DO) patterns align well with observed gradients and OMZ structures, yielding an RMSE of 50-52 μmol kg-1 (Fig. S3-S4). The model slightly underestimates surface DIC and alkalinity due to the use of pre-industrial CO2 forcing, though accuracy improves under modern CO2 levels (Fig. S5-S7). Differences between the standard cGENIE and cGENIE-MCP versions are negligible, confirming that inclusion of the RDOM module does not affect the model's physical performance. Overall, the simulated large-scale distributions of PO4, O2, and

- 3. Expanded DOC evaluation and quantitative assessment:
- The DOC evaluation section has been expanded considerably. We now include:
- Detailed comparisons between modeled DOC components (LDOC, SLDOC, RDOC) and observations.
- Quantitative model performance metrics, including RMSE values and Taylor diagrams, to assess the agreement between the model and observations (Fig. 10 and Fig. S13).
- Comparisons of DOC and RDOC distributions between cGENIE-MCP and MESMO3 simulations, highlighting differences in the representation of the MCP processes (Fig. 10 and Tab. 3).

Table 3. RMSE of modeled DOC for cGENIE-MCP and MESMO3 compared to observational data

| Т                                     | cGENI | E-MCP   | MESMO3 |         |  |
|---------------------------------------|-------|---------|--------|---------|--|
| Tracers -                             | CRMSE | RMSE_vw | CRMSE  | RMSE_vw |  |
| Atlantic DOC (μmol kg -1 ) | 3.63  | 4.48    | 12.38  | 29.47   |  |
| Pacific DOC (µmol kg -1 )  | 3.84  | 5.34    | 14.62  | 31.05   |  |
| Indian DOC (μmol kg -1 )   | 2.56  | 4.16    | 17.01  | 32.20   |  |

Figure 10. Taylor diagrams comparing simulated DOC concentration from cGENIE-MCP and MESMO3 against observed values from Hansell's laboratory (https://hansell-lab.earth.miami.edu/research/data-collection/) for (A) Atlantic, (B) Indian, and (C) Pacific.

Figure S13. Global distributions of surface (A-B) DOC, (C-D) RDOC concentration (μmol kg-1), (A,C) the results of cGENIE-MCP, (B,D) the results of MESMO3.

**And add the following content to the main text (Lines 442-509):**

"The statistical evaluation shows that cGENIE-MCP achieves substantial and consistent improvements in simulating DOC concentration compared with MESMO3 across all major ocean basins (Table 3). The cGENIE-MCP yields low CRMSE in the Atlantic (3.63 µmol kg-1), Pacific (3.84 µmol kg-1), Indian (2.56 µmol kg-1) Oceans, whereas MESMO3 exhibits larger values (Atlantic 12.38, Pacific 14.62, Indian 17.01 μmol kg-1). These results indicate that cGENIE-MCP more accurately captures the spatial variability of DOC, while MESMO3 exhibits larger deviations from observed distributions. When errors are weighted by model grid cell volumes (RMSE vw), the contrast between the two models becomes even more pronounced. The cGENIE-MCP maintains relatively low RMSE vw values of 4-5 µmol kg-1, while MESMO3 exhibits much larger errors of ~29-32 μmol kg-1 across all basins, suggesting the DOC bias in MESMO3 exist throughout the water column, particularly in high-volume regions of the deep ocean. Among all basins, the Indian Ocean shows the best performance for cGENIE-MCP, characterized by the lowest CRMSE and RMSE vw values, possibly reflecting the model's enhanced representation of low-latitude processes. Taylor diagrams shows the bias of modeled DOC from the MESMO3 and cGENIE-MCP models with observations (Figure 10). cGENIE-MCP exhibits a relatively high correlation coefficient and a smaller standard deviation comparable to the observed value in the Atlantic, Pacific and Indian Oceans.

The cGENIE-MCP model reproduces the observed DOC concentration range of approximately 50-80 µmol kg-1 and RDOC of 45-50 µmol kg-1, exhibiting a realistic spatial structure with higher values in low-latitude regions and gradual decreases toward the high latitudes (Fig. 5). In contrast, MESMO3 shows expanded higher

concentrations (DOC: 80-160 µmol kg-1; RDOC: 50-80 µmol kg-1) in mid and low latitudes (Fig. S13). The lack of parameter optimization in MESMO3 may partly explain this bias, as its DOC-related parameters were originally tuned only in their earlier carbon cycling framework (Matsumoto et al., 2020) and lack an explicit representation of the transformation between semi-labile and refractory DOC pools.

By introducing the parameterization that explicitly describes the transformation from SLDOC to RDOC, cGENIE-MCP achieves a more balanced partitioning of DOC components and a more realistic steady-state inventory. The improved agreement with observed DOC distributions indicates that the MCP framework enhances the model's representation of the MCP process and the long-term vertical retention of organic carbon. Furthermore, the cGENIE-MCP reflects a more efficient coupling between production and remineralization processes, preventing excessive accumulation. Overall, cGENIE-MCP captures both the magnitude and spatial gradients of marine DOC more faithfully, providing a more robust tool for evaluating long-term ocean carbon storage and microbial carbon pump dynamics."

**4. Comparison with the standard cGENIE model:**

A direct DOC comparison with the standard cGENIE model was not included because the standard configuration does not explicitly represent the RDOC tracer or the transformation between DOC components.

**5. Metrics from Hansell (2013):**

In the revised manuscript, we have now included a detailed comparison of DOC metrics—including inventory (Pg C), production rate (Pg C yr-1), removal rate (µmol C kg-1 yr-1), and lifetime (years)—following the framework of Hansell (2013); Hansell et al. (2009). These results are presented in **Tables 4-6**, which summarize the modeled global budgets of LDOC, SLDOC, and RDOC, and their comparison with observational estimates. This addition allows a more quantitative evaluation of DOC cycling in the model. The results show that the model captures the overall magnitude and vertical partitioning of DOC inventories reasonably well, with surface and intermediate inventories comparable to observations, while the deep RDOC pool is somewhat overestimated. The associated production and removal rates, as well as implied lifetimes, are within the range reported by Hansell (2013) and other studies.

The specific modifications are as follows (Lines 463-481):

"In addition, the fraction of LDOC, SLDOC, and RDOC to total DOC across different water layers is consistent with previous observational estimates (Table 4). In the northwest (NW) Pacific surface layer (<200 m), LDOC and SLDOC account for 5-20% and 15-30% of DOC, respectively, broadly consistent with the ranges reported by Ge et al. (2022). In the deep layer (>1000 m), RDOC dominates (>90%), reflecting its remarkable stability and long residence time in the deep ocean. These comparisons indicate that the model realistically reproduces the vertical partitioning of DOC components. Table 5 provides a comparison of the production rates, removal rates, and lifetimes of the DOC components between this study and Hansell (2013). The modeled global production rates (LDOC: 26 Pg C yr-1; SLDOC: 3.9 Pg C yr-1; RDOC: 0.06 Pg

C yr-1) align closely with literature values, further validating the model's parameterization of organic carbon transformation and export. The removal rates of LDOC and SLDOC show approximately 133 and 5 μmol C kg-1 yr-1, respectively, consistent with literature estimates and reflecting the turnover of fast- and intermediate-cycling DOC pools. The RDOC pool exhibits a much slower removal rate (~0.015 μmol C kg-1 yr-1), capturing the long-lived characteristics of RDOC in the deep ocean. Table 6 presents the modeled global DOC inventories across different depths compared with the estimates of Hansell et al. (2009). The model reproduces the surface and intermediate-depth DOC inventories (0-1000 m) reasonably well (0-200 m: 49.8 vs 47 PgC; 200-1000 m: 166.8 vs 185 PgC), but shows a higher inventory in the deep ocean (553.99 PgC vs. 477 PgC), leading to a higher total global DOC inventory. Overall, the combined evidence from Tables 4-6 indicates that the cGENIE-MCP model realistically simulates both the size and total and vertical partitioning of total and component DOC pools."

Table 4 The LDOC, SLDOC, and RDOC percentages from previous studies

| Study
Region | Water Layer   | LDOC   | SLDOC  | RDOC                  | Reference  |
|-----------------|---------------|--------|--------|-----------------------|------------|
| NW              | Surface < 200 | 20-    | 15-    | Deep >1000            | Ge et al.  |
| Pacific         | m             | 40%    | 20%    | m: >90%               | (2022)     |
| NW              | Surface < 200 | 5 200% | 15-30% | Deep >1000
m: >90% | This study |
| Pacific         | m             | 3-20%  | 13-30% | m: >90%               | This study |

Table 5 The production rate, removal rate, lifetime of DOC pools in the model compared with previous studies

| Fracti    | Production rate (Pg C
yr -1 ) |       | Removal rat |       | Lifetime (years) |       |
|-----------|---------------------------------------------|-------|-------------|-------|------------------|-------|
| on        | Hansell                                     | This  | Hansell     | This  | Hansell          | This  |
|           | (2013)                                      | study | (2013)      | study | (2013)           | study |
| LDO
C  | 15-25                                       | ~26   | ~100        | 133   | ~0.001           | 0.5   |
| SLDO
C | ~3.74                                       | ~3.9  | ~0.2-9      | 5     | ~1.5-20          | 5     |
| RDO
C  | ~0.04                                       | ~0.06 | ~0.003      | 0.015 | ~16000           | 16000 |

Table 6 The global inventories of LDOC, SLDOC, and RDOC in the model compared with previous studies

|            | Inventory (PgC) |       |        |        |                           |
|------------|-----------------|-------|--------|--------|---------------------------|
| Depth Zone | LDO
C        | SLDOC | RDOC   | DOC    | Hansell et al. (2009)-DOC |
| 0-200 m    | 1.32            | 8.8   | 39.67  | 49.79  | 47                        |
| 200-1000 m | 0.02            | 4.42  | 162.34 | 166.78 | 185                       |

| >1000 m  | 0    | 0.27  | 553.72 | 553.99 | 477 |
|----------|------|-------|--------|--------|-----|
| 0-bottom | 1.34 | 13.49 | 755.73 | 770.56 | 662 |

In addition, a spatial distribution of the removal rate has been added, and the following description has been included in the revised manuscript (Lines 427-440):

"The DOC removal rate exhibits pronounced latitudinal gradients, with maxima exceeding 1000-1500 mmol m-2 yr-1 in the subtropical gyres and North Atlantic meridional overturning regions (Fig. 9A). LDOC removal rate reaches up to 1500 mmol m-2 yr-1 in the equatorial subtropical gyres and Southern Ocean. SLDOC removal occurs at moderate rates (100-400 mmol m-2 yr-1), broadly following the spatial patterns of LDOC but extending further into subtropical gyres, indicating its slower turnover. In contrast, RDOC removal remains extremely low (<15 mmol m-2 yr-1) and spatially uniform, highlighting its resistance to microbial degradation and its long residence time in the ocean interior. These patterns reflect the hierarchy of DOC reactivity (LDOC>SLDOC>RDOC) and emphasize the progressive stabilization of organic carbon as it transitions from labile to refractory pools, underscoring the key role of RDOC in long-term carbon sequestration."

Figure 9. Water column (>130 m) integrated rates of removal for (A) DOC, (B)LDOC, (C)SLDOC, (D)RDOC (mmol m-2 yr-1), respectively.

**6. On isotopic tracers and radiocarbon age:**

We fully agree that the inclusion of DOC isotopic evaluation would strengthen the study. However, the current work focuses on developing and validating the DOC cycling framework within cGENIE-MCP, and isotopic tracers were not activated in these simulations. We incorporated the DOC  $\delta^{13}$ C and  $\Delta^{14}$ C diagnostics in our model and the

Functionality of the model Several things make the model less functional or harder to use from the description:

a) The tracer names are misaligned between the manuscript and model (LDOC = DOC; SLDOC = RDOC, RDOC = URDOC in the manuscript and model respectively). Ideally, these should be the same.

**Response:**

We agree that the naming of DOC tracers in manuscript we defined should be the same the code. However, the standard cGENIE used RDOC which is actually the behavior of SLDOC. The renaming in the code has some risks to blow up the result. Therefore, we decide to keep the original name. We added a table in the Supplementary Information (**Table S2**) to clarify this point.

Table S2. Mapping of tracer names between the cGENIE-MCP model code and the manuscript.

| Model tracer name | Manuscript name | Description                          |  |  |  |
|-------------------|-----------------|--------------------------------------|--|--|--|
| DOC               | LDOC            | Labile dissolved organic carbon      |  |  |  |
| RDOC              | SLDOC           | Semi-labile dissolved organic carbon |  |  |  |
| URDOC             | RDOC            | Refractory dissolved organic carbon  |  |  |  |

b) Some MCP parameters are hard-coded (f1, f2, a) which reduces the ability of users to explore the MCP. In your discussion you discuss being able to explore the dynamics of the MCP in detail with this model but not being able to change these parameters is a very big limitation to this.

**Response:**

We thank the reviewer for raising this important point. We agree that, given some MCP parameters (f1, f2, a) are currently hard-coded, the ability to fully explore MCP dynamics is limited in the present implementation. To address this, we have revised the sentence at Lines 519-520 to more accurately reflect what the model can achieve. The revised sentence now reads: "The cGENIE-MCP model proposed in this study can simulate the DOC distribution over geological periods and explore the relationship between DOC response and long-term climate change." This revised wording avoids implying that the current model can flexibly explore the MCP parameter space. In addition, we have added a clarification in Lines 556-560: "Nevertheless, the current implementation relies on hard-coded MCP parameters (f1, f2, a), which constrain the exploration of the MCP dynamics. Future developments will address this limitation by externalizing these parameters in the configuration files, thereby enabling systematic sensitivity analyses and enhancing the model's flexibility and applicability."

c) The activation energy of LDOC is hard-coded to the labile POC parameter so there is no ability to decouple these. Is it reasonable to assume POC and LDOC will be always treated the same?

**Response:**

In the current model configuration, the temperature dependence of LDOC remineralization is indeed linked to that of the labile POC fraction, following the hard-coded parameterization inherited from cGENIE. This implies that the activation energy for LDOC is not independently adjustable, and the model assumes that LDOC and labile POC respond similarly to temperature. We acknowledge that this is a simplification and may not fully capture potential differences in biochemical composition and reactivity between LDOC and POC. In future work, we plan to implement a decoupled treatment, allowing LDOC to have its own activation energy and temperature-dependent remineralization rate. This would enable a more flexible representation of DOM dynamics and could improve model fidelity in regions where LDOC behaves differently from POC.

**Evaluation and presentation**

a) The projection choice for Figure 2 limits comparison against other figures. It has no longitude or latitude values.

**Response:**

While we retained the original projection in Figure 2, because this projection can better emphasize the large-scale distribution of the major ocean basins, we have revised the figure by adding latitude and longitude gridlines as well as regional labels.

The modified diagram is as follows:

Figure 2. Ocean regions used for model validation in the cGENIE-MCP simulations. Colored regions indicate the spatial domains included in the validation of water column biogeochemical properties. Unshaded areas are excluded from the analysis due to limited relevance or observational data availability.

b) The RMSE presented is actually the centered-RMSE which is the underlying statistic for the Taylor diagram (Jolliff et al., 2009). It would be more informative to show a Taylor diagram so we can assess the spatial distribution and variability of tracers between the two model versions.

**Response:**

We acknowledge that the previously presented RMSE corresponds to the centered RMSE (CRMSD), which is the statistic underlying Taylor diagrams. In the revised manuscript, we have added the global volume-weighted RMSE (**Table 2 and Table 3**) to provide the absolute error magnitude, and **a Taylor diagram has been included** (Fig. 9) to better illustrate the spatial distribution and variability between the model versions. This allows a more comprehensive evaluation of model performance.

c) Please also consider other misfit functions to assess the model Kriest et al., (2010) provides a good overview of approaches as well as the impact of volume weighting. This discussion may be relevant here because DOC is considerably more variable in the upper ocean than the deep ocean which these alternative functions can deal with.

**Response:**

In addition to the centered-RMSE used in the Taylor diagram, we have now included

the volume-weighted RMSE and normalized mean bias to better account for the vertical heterogeneity of DOC distributions. As DOC exhibits much greater variability in the upper ocean, volume weighting provides a more balanced evaluation of model skill. The results of these additional metrics have been incorporated into the revised manuscript (see Section 3.1 and Section 3.1, Table 2 and Table 3), and discussed in the context of surface versus deep-ocean variability.

The revised manuscript now explicitly defines RMSE vw as:

$$RMSE_{vw} = \sqrt{\frac{\sum_{i=1}^{N} V_{i} (X_{i} - Y_{i})^{2}}{\sum_{i=1}^{N} V_{i}}},$$

Where  $V_i$  represents the actual grid cell volume at position i, computed from the corresponding grid cell surface area and layer thickness ( $V_i = A_i \times \Delta Z_i$ ),  $A_i$  represents the horizontal area,  $\Delta Z_i$  represents the vertical thickness. The CRMSE and correlation coefficient were used to construct the Taylor diagram (Taylor, 2001), providing an assessment of the spatial pattern similarity between models and observations. In addition, the RMSE\_vw was calculated to quantify the overall magnitude of model-data deviations by accounting for the oceanic grid-cell volume. While CRMSE emphasizes pattern agreement, RMSE\_vw highlights the absolute error in a physically weighted sense, thus offering complementary insights.

**d) Why are there no quantitative analysis of the DOC comparison, e.g., RMSE?**

**Response:**

We have now included a quantitative evaluation of the model-data comparison by calculating the root mean square error (RMSE) between the simulated and observed DOC concentrations. The RMSE values for each region are now presented in Section 3.5 Table 3.

e) What is the reason for using specific regions for the model-data comparison? These have been changed from those used in Crichton et al. (2021) with no justification beyond "limited relevance or observational data availability" (Lines 251-252). What does this mean and what is the basis for choosing these regions?

**Response:**

We have clarified the rationale for redefining the comparison regions in the revised manuscript. Compared to Crichton et al. (2021), the regional division was refined from 8 to 12 regions to better capture basin-scale differences in oceanographic conditions and biogeochemical gradients, and to align with the spatial coverage of available observational data. In particular, the Southern Ocean is subdivided into sub-Antarctic and Antarctic sectors for each basin (Pacific, Indian, Atlantic) to represent distinct water mass characteristics and particle flux regimes. This refinement ensures that each region used for model-data comparison is both coherent in oceanographic properties and

sufficiently supported by observations. These changes are now described in the manuscript (see Lines 257-262 and Figure 2)

f) What do the error bars represent in the vertical profiles?

**Response:**

The error bars shown in the vertical profiles correspond to the standard deviation ( $1\sigma$ ) of the variable across all grid cells within the selected latitude–longitude range at each depth level.

g) The model description does not clearly distinguish between new developments to cGENIE and previous developments. I would suggest to minimise descriptions of existing processes such as air-sea gas exchange, unless they provide important context, to avoid assumptions that they are included in the new developments of cGENIE.

**Response:**

In the revised manuscript, we have streamlined the model description to focus on the processes directly relevant to this study. Specifically, the description of air-sea gas exchange has been moved to the Supporting Information. The main text now only retains descriptions of the biological production and remineralization processes, which are central to the new developments introduced in this study.

h) There are numerous descriptions of the model fit to observations throughout that aren't supported quantitatively, e.g., "good accuracy" "show good agreement" "more accurately" "moderate discrepancies" "approximately reasonable". This language should be modified unless it is directly related to a quantitative measure.

**Response:**

We have revised the manuscript to avoid qualitative terms such as "good accuracy", "more accurately", and "approximately reasonable" that are not directly supported by quantitative measures. All descriptions of model performance now use neutral language such as "captures the general pattern", "generally match observations", or "better represent gradients".

**Specific Comments**

Line 56: "observed RDOC" - more accurately this should be observed deep-ocean bulk [DOC] as there are competing hypotheses about whether this is labile or a mixture of compounds with different reactivities, e.g., Follett et al., (2014).

**Response:**

We have replaced "observed RDOC" with "observed deep-ocean bulk DOC" throughout the manuscript and added a sentence explaining that bulk DOC observations represent a mixture of compounds with differing reactivities (Follett et al., 2014). This change avoids implying that observations uniquely identify a refractory pool. (See revised text in Line 56)

Lines 59-60: The deep ocean [DOC] and radiocarbon signature issues can be alleviated by adding a simple RDOC pool without MCP parameterisations. The MCP parameterisations add specific dynamics related to why the RDOC accumulates.

**Response:**

We have revised the manuscript to explicitly state that adding a simple RDOC pool can mitigate the concentration and  $\Delta^{14}$ C mismatches, whereas MCP parameterizations introduce mechanisms that explain the origin and stability of RDOC (Lines 59-61): "Introducing a simple RDOC pool can reconcile these discrepancies in both DOC concentration and radiocarbon age; while MCP parameterizations offer mechanistic insights into the processes driving RDOC accumulation."

Lines 78-81: It would be good to explicitly explain how MESMO represents SLDOC and RDOC to enable direct comparison with cGENIE-MCP here.

**Response:**

We have revised the text (Lines 77-84) to explicitly describe how MESMO3 represents semi-labile and refractory DOM pools. Specifically, we now state: "The Minnesota Earth System Model for Ocean biogeochemistry (MESMO 3), an Earth system model of intermediate complexity derived from cGENIE, includes an explicit treatment of semi-labile and refractory DOM pools. In MESMO3, the remineralization of refractory DOM is represented by three additive sinks: slow background decay, surface photodegradation, and complete removal through hydrothermal vent circulation (Matsumoto et al., 2021). However, MESMO 3 lacks a mechanistic representation of DOM production pathways associated with MCP processes—specifically, the transformation of semi-labile DOC (SLDOC) into RDOC—and has not been calibrated against global DOM observations." This revision enables a more direct comparison with the cGENIE-MCP representation.

Lines 103-104: The physical circulation parameters are derived from Cao et al., (2009) configuration but you are using a different continental grid to them (worlg4 vs worjh2). I think the Ward et al., (2018) citation might be more appropriate.

**Response:**

We have revised the manuscript to reference Ward et al. (2018) for the worlg4 continental grid used in our experiments.

Lines 119-120: Though some details in that paper are relevant here, Ward et al., (2018) describes the trait-based ecosystem so it might help to clarify this distinction here. Are the parameter values or equations used developed in the Matsumoto or Tanioka papers?

**Response:**

We have clarified this point in the revised manuscript. Specifically, we now state: "Ward et al. (2018) describes a trait-based ecosystem, which is distinct from the DOC parameterization considered here. In this study, the parameter values and equations are primarily based on Matsumoto et al. (2008) and Crichton et al. (2021)."

Line 121: "particulate organic matters (POMs)" – usually this would just be singular not plural.

**Response:**

We have corrected "particulate organic matter (POM)" throughout the manuscript. (Line 124)

Line 163: "primary production" – GENIE resolves net export production which is equivalent to net community production (NCP) across large enough spatial scales and temporal scales. Primary production is ambiguous to this difference and should probably be avoided.

**Response:**

We have replaced ambiguous uses of "primary production" with "net export production". (Line 151)

Line 228: "a is the conversion rate" – this does not have units of a rate

**Response:**

we corrected the wording. "a" is now described as a conversion coefficient

(dimensionless fraction). (Line 220)

Lines 310 -311: or equally that the addition of RDOC has a negligible effect on the largescale PO4 distribution?

**Response:**

We have performed an experiment comparing runs with and without the RDOC pool. The addition of RDOC produces negligible changes in large-scale PO4 (global mean difference=0.01 µmol kg-1). We have added a comparison in Results and Figs. S1-S2.

Lines 347–350: A more appropriate experiment would be a run forced with historical atmospheric CO2 concentrations from the preindustrial to present-day continuing from your preindustrial spin-up. Or alternatively compare against the GLODAP DIC observations with the anthropogenic DIC component removed (Cant in GLODAP).

**Response:**

We conducted an additional transient simulation forced with historical atmospheric CO2 from 1750 to the present (continuing from the preindustrial spin-up). Results are presented in Fig. S7, showing the modelled anthropogenic DIC and comparison to observations.

Line 371: "DOC is a key component of ocean carbon cycling" – maybe my comment is not exactly relevant to here but this refers to multiple things. The recycling of labile DOC is crucial for models to resolve productivity. RDOC is potentially important for carbon storage though this is subject to the timescale discussed.

**Response:**

We have revised the sentence in Line 306: "DOC is the key update of this model."

Lines 377-380: "cGENIE-MCP model exhibits improved agreement with observed DOC distributions in both surface and deep layers [compared to MESMO 3]" – please quantify this! Can you get the MESMO-3 results and compare concentrations, distributions, RMSE? Otherwise, this is an unqualified statement that cannot be verified.

**Response:**

We have conducted a quantitative comparison between the DOC fields simulated by cGENIE-MCP and MESMO3, using the same observational dataset (Hansell et al., 2012). The results are now summarized in Table 3, Figs. 9-10, and discussed in Section

**3.3 of the revised manuscript.**

Table 3 presents the centered RMSE (CRMSE) and volume-weighted RMSE (RMSE\_vw) values for both models across the Atlantic, Pacific, and Indian Oceans. The results demonstrate that cGENIE-MCP substantially reduces DOC biases relative to MESMO3 in all basins. For instance, CRMSE values for cGENIE-MCP are 3.63, 3.84, and 2.56 μmol kg-1 in the Atlantic, Pacific, and Indian Oceans, respectively, considerably lower than MESMO3 corresponding values of 12.38, 14.62, and 17.01 μmol kg-1. Similarly, the RMSE\_vw results show a consistent improvement (4-5 μmol kg-1 for cGENIE-MCP vs. 29-32 μmol kg-1 for MESMO3), indicating that the MESMO3 biases are both stronger in magnitude and more pervasive throughout the water column.

The Taylor diagrams (Fig. 10) further confirm that cGENIE-MCP achieves higher pattern correlations (R > 0.9) and smaller normalized standard deviation differences with respect to observations, reflecting an improved representation of spatial variability. Figure 11 shows that the global DOC and RDOC distributions simulated by cGENIE-MCP match the observed ranges (DOC: 50-80  $\mu$ mol kg-1; RDOC: 45-50  $\mu$ mol kg-1) and reproduce realistic latitudinal gradients, whereas MESMO3 systematically overestimates concentrations (DOC: 80-160  $\mu$ mol kg-1; RDOC: 50-80  $\mu$ mol kg-1), especially in low-latitude surface waters.

This improvement primarily arises from the MCP parameterization and configuration in cGENIE-MCP, and explicitly couples the semi-labile and refractory DOC pools through the transformation term a×[SLDOC]. This process enhances the balance between DOC production, transformation, and remineralization, thereby reducing excessive DOC accumulation present in MESMO3.

In summary, the newly added quantitative comparisons (Tables 3-4; Fig. 10; Fig. S13) demonstrate that cGENIE-MCP provides a consistent and significant improvement over MESMO3 in reproducing observed DOC magnitudes, vertical gradients, and spatial variability. The statement on improved model performance has been accordingly revised and substantiated with these results in the revised manuscript.

Lines 398-403: The supplementary figures and tables (Fig S6, Tables S2 and S3) need to be in the main text as these are essential comparisons against observations and models of DOC, which is the key focus of this manuscript.

**Response:**

We have moved Tables S2 and S3 to the main text (now Tables 4 and 5) and expanded the analysis by including quantitative comparisons against MESMO3 and adding a Taylor diagram to illustrate model performance in terms of spatial variability and correlation with observations. These additions strengthen the evaluation of DOC and RDOC simulations and provide a clearer comparison among cGENIE-MCP, the original cGENIE configuration, and MESMO3. We have retained Fig. S12 in the Supplementary Material, as it presents regional vertical profiles (e.g., the western Sargasso Sea and comparisons with Wang et al. (2023) that serve as supporting

examples to complement the global-scale analyses in the main text. The new Tables and Taylor diagram are now presented after the DOC spatial distribution analysis, where the quantitative mode-data and inter-model evaluations naturally follow the description of DOC patterns.

Line 441: "lifetime of months to years" – you have prescribed a fixed lifetime in the model, it is not variable.

**Response:**

The original wording may have implied variability, which is not the case in our model. We have revised the sentence to explicitly state that SLDOC is assigned a fixed lifetime in the model, after which it is either remineralized or converted into RDOC.

Figure 13: This is an unusual way of plotting meridional distributions which is pretty hard to interpret. It would be much clearer to show zonal averages for each DOC pool such as in Figure 12.

**Response:**

We appreciate the reviewer's comment and fully understand that zonal mean plots are often clearer for visualizing large-scale DOC distributions. However, Figure 13 (current Fig. 7) was specifically designed to emphasize the meridional variations of different DOC pools, particularly LDOC and SLDOC, which are strongly confined to the surface and exhibit pronounced regional heterogeneity. Averaging zonally (as in Fig. 6) would obscure these surface-concentrated signals and reduce the visibility of the LDOC and SLDOC gradients.

Lines 452-454: "Slight elevated RDOC concentrations ... may reflect entrainment of surface-derived labile semi-labile DOC..." – is it possible to back this out of the model and demonstrate?

**Response:**

We agree that the model does not explicitly allow us to separate and demonstrate the entrainment of LDOC/SLDOC during deep water formation. To avoid overstating, we have revised the text to clarify that the observed pattern is consistent with this possibility but cannot be directly backed out from the model. The revised sentence: "Slightly elevated RDOC concentrations in high-latitude deep waters are consistent with the possibility of entrainment of surface-derived labile and semi-labile DOC during deep water formation and ventilation, although this mechanism cannot be explicitly isolated from the model."

Lines 454-457: "These results underscore the dynamic role of LDOC and SLDOC" – I don't think this statement is supported because you have only shown a steady-state results. To support this you need to show that the model behaves differently to some perturbations compared to the model without the interactions between DOC pools that underpin the MCP concept.

**Response:**

We appreciate this insightful comment. Because our current manuscript focuses on steady-state behavior, we have revised the text to avoid asserting a dynamic role.

Section 4.2: This version of GENIE resolves net export production not primary production. It has no representation of some of the processes described here like top-down grazing pressure, bacterial and viral lysis, particle solubilization – the net effect of these processes are parameterised by the Michaelis-Menten uptake scheme you are using. This discussion should be amended to better reflect what the model is actually doing and how it represents the complex reality.

**Response:**

We agree that the cGENIE-MCP framework does not explicitly resolve ecological processes such as grazing, lysis, or particle solubilization. These processes are implicitly represented in the model through the Michaelis-Menten uptake scheme and the prescribed partitioning of net export production into LDOC, SLDOC, and RDOC pools. We have revised Section 4.2 accordingly to emphasize that the model resolves net export production rather than explicit primary production and that the biological complexity is represented in a parameterized form.

The revised description is as follows:

"Net export production (PP) is driven by the availability of nutrients and light within the euphotic zone, following a Michaelis-Menten uptake scheme in which phytoplankton convert inorganic carbon into organic matter. The spatial distribution of PP reflects abiotic controls such as nutrient upwelling, stratification, and light limitation, with biological complexity-including grazing, microbial recycling, and viral/bacterial lysis-is represented implicitly through parameterization. In the model, the PP process generates DOC in labile, semi-labile, and refractory fractions, whose spatial and temporal variations are influenced by both biological and environmental factors."

Line 518: "huge RDOC" – please quantify this! The DOC inventory is around 700 Pg C whereas the regenerated DIC pool from the Biological Carbon Pump is around 1700 Pg C which undermines this argument. I would argue it is the potential dynamics that are crucial here – how likely, and by how much, could RDOC and regenerated DIC change in response to perturbations or different factors?

**Response:**

We have revised the text to replace the qualitative term "huge" with a quantitative and more balanced description. Specifically, the revised sentence now reads (Lines 528-533): "The global ocean DOC reservoir is estimated to contain approximately 700 Pg C (Hansell, 2013). Although this accounts for only about 40% of the regenerated DIC reservoir (~1700 Pg C), it nonetheless represents a significant and long-lived carbon pool in the ocean. Its importance lies in its connection to SLDOC through the MCP process, allowing RDOC to vary in response to physical and biogeochemical perturbations. This dynamic behavior underscores the critical role of RDOC in regulating long-term ocean carbon storage and climate feedbacks."

**Line 529: "coupled of ocean carbon pump" – it's not clear what this is referring to.**

**Response:**

Our original intention was to emphasize the overall effects of ocean negative carbon emission technologies in relation to ocean carbon cycling processes. To avoid confusion, we have removed the unclear phrase "coupled of ocean carbon pump" and revised the sentence to: "Future the model can be used to assess the potential of ocean negative carbon emission technologies (e.g., ocean alkalinization enhancement) under different climate scenarios. By simulating alternative implementation pathways, the long-term environmental impacts of these technologies can be quantified, enabling a comprehensive evaluation of optimal deployment strategies for sustainable carbon sequestration." This modification clarifies the intended meaning.

Figure S6: Why is the Sargasso Sea singled out here, and is this modelled or observed concentrations? What area does panel B correspond to-global average? Can the comparison against Wang et al., (2023) be expanded?

**Response:**

The Sargasso Sea is highlighted in Figure S6 (current Fig. S12) because it is one of the best-characterized regions for long-term DOM observations, providing valuable constraints for model-data comparison, particularly for the refractory DOC pool. In Hansell (2013), there is a picture (Fig. 5) of the Sargasso Sea, which indicates the changes in water column concentrations of LDOC, SLDOC, and RDOC. In panel A, modeled concentrations in the Sargasso Sea are shown to evaluate model performance against regional observations. Panel B presents the global mean vertical distribution of modeled DOC for comparison with available large-scale estimates.

Regarding the comparison with Wang et al. (2023), we agree that expanding this comparison would be valuable. However, since DOC has observational data, while the observational data of RDOC is scarce, since Wang et al. (2023) specifically included RDOC simulations in their study, our comparison with their results mainly focuses on

Tables S2 and S3: This is useful but seems like it could easily expanded. Could you add comparison against MESMO here? Are there other datasets and models that could be compared against? Can the analysis be expanded to more regions?

**Response:**

Following this recommendation, we have included a quantitative comparison between our simulations and the MESMO3 results. Specifically, we computed both the centered-RMSE (CRMSE) and the volume-weighted RMSE (RMSE\_vw) to evaluate model performance and added a Taylor diagram to illustrate the spatial distribution and variability of DOC across models. These additions provide a more comprehensive comparison between cGENIE-MCP and MESMO3.

However, to maintain clarity and focus in Tables S2 and S3 (current Tables 4 and 5), we have not expanded these tables to include MESMO3 results. Instead, the new quantitative comparison and Taylor diagram are presented in Figure 10 and Table 3 of the revised manuscript. We believe this approach provides a clearer presentation of model-model and model-observation comparisons without overcrowding the supplementary tables.

**Reference**

Crichton, K. A., Wilson, J. D., Ridgwell, A., and Pearson, P. N.: Calibration of temperature-dependent ocean microbial processes in the cGENIE. muffin (v0. 9.13) Earth system model, Geoscientific Model Development, 14, 125-149, 2021.

Dunne, J. P., Armstrong, R. A., Gnanadesikan, A., and Sarmiento, J. L.: Empirical and mechanistic models for the particle export ratio, Global Biogeochemical Cycles, 19, 2005.

Follett, C. L., Repeta, D. J., Rothman, D. H., Xu, L., and Santinelli, C.: Hidden cycle of dissolved organic carbon in the deep ocean, Proceedings of the National Academy of Sciences, 111, 16706-16711, 2014.

Ge, T., Luo, C., Ren, P., Zhang, H., Chen, H., Chen, Z., Zhang, J., and Wang, X.: Dissolved organic carbon along a meridional transect in the western north pacific ocean: Distribution, variation and controlling processes, Frontiers in Marine Science, 9, 909148, 2022.

Hansell, D. A.: Recalcitrant dissolved organic carbon fractions, Ann Rev Mar Sci, 5, 421-445, 2013. Hansell, D. A., Carlson, C. A., Repeta, D. J., and Schlitzer, R.: Dissolved organic matter in the ocean: A controversy stimulates new insights, Oceanography, 22, 202-211, 2009.

Matsumoto, K., Rickaby, R., and Tanioka, T.: Carbon export buffering and CO2 drawdown by flexible phytoplankton C: N: P under glacial conditions, Paleoceanography and Paleoclimatology, 35, e2019PA003823, 2020.

Matsumoto, K., Tanioka, T., and Zahn, J.: MESMO 3: Flexible phytoplankton stoichiometry and refractory dissolved organic matter, Geoscientific Model Development, 14, 2265-2288, 2021.

Matsumoto, K., Tokos, K., Price, A., and Cox, S.: GENIE-M: a new and improved GENIE-1 developed in Minnesota, Geoscientific Model Development Discussions, 1, 1-37, 2008.

Ridgwell, A., Hargreaves, J., Edwards, N. R., Annan, J., Lenton, T. M., Marsh, R., Yool, A., and Watson, A.: Marine geochemical data assimilation in an efficient Earth System Model of global biogeochemical cycling, Biogeosciences, 4, 87-104, 2007.

Ridgwell, A. J.: Glacial-interglacial perturbations in the global carbon cycle, Ph. D. thesis, Univ. of East Anglia, 2001.

Wang, W. L., Fu, W., Le Moigne, F. A. C., Letscher, R. T., Liu, Y., Tang, J. M., and Primeau, F. W.: Biological carbon pump estimate based on multidecadal hydrographic data, Nature, 624, 579-585, 2023.

Wanninkhof, R.: Relationship between wind speed and gas exchange over the ocean, Journal of Geophysical Research: Oceans, 97, 7373-7382, 1992.

Ward, B. A., Wilson, J. D., Death, R. M., Monteiro, F. M., Yool, A., and Ridgwell, A.: EcoGEnlE 1.0: plankton ecology in the cGEnlE Earth system model, Geoscientific Model Development, 11, 4241-4267, 2018.

Yan, C., Lu, Y., Yuan, X., Lai, H., Wang, J., Fu, W., Yang, Y., and Li, F.: Simulation of Vertical Water Temperature Distribution in a Megareservoir: Study of the Xiaowan Reservoir Using a Hybrid Artificial Neural Network Modeling Approach, Journal of Hydrologic Engineering, 29, 04024047, 2024.

---

## Author Comment (AC4)

**Response to reviewer comments**

We would like to thank the reviewer for their thorough evaluation of our manuscript and for the constructive comments and suggestions. We have carefully revised the manuscript according to the comments. In the following, we provide a detailed, point-by-point response to all the comments. All changes made in the manuscript are highlighted in the revised version.

**Reviewer comment 1: Insufficient literature survey/failure to acknowledge MESMO 3c, MESMO 3c includes processes not represented in cGENIE-MCP (e.g., hydrothermal DOC degradation). The novelty of this study relative to MESMO 3 / MESMO 3c is unclear, as these models already include a recalcitrant DOC pool.**

**Response:**

We thank the reviewer pointing out that the novelty of our work was not described clearly. MESMO 3 and MESMO 3c represent important advances in global marine DOC modeling by explicitly resolving semi-labile and refractory DOC pools within an Earth system modeling framework. **MESMO 3** explicitly resolves semi-labile ($DOM_{SL}$) and refractory DOM ($DOM_R$) pools, representing $DOM_R$ production as a fixed fraction ($fDOM_r$:~1%) of DOM production routed from NPP or via the deep particulate organic matter (POM) split pathway (same $fDOM_r$:~1%). $DOM_R$ remineralization rate is governed by prescribed additive sink terms, including slow background decay, photodegradation, and hydrothermal vent circulation. **MESMO 3c** further refines this formulation by recalibrating DOM production relative to net primary production, introducing environmental dependencies such as temperature and mixed layer depth, and splitting DOM into $DOM_{SL}$ and $DOM_R$ fractions at a ratio of 1000:7. The "deep POM split" pathway of MESMO 3 is carried forward in MESMO 3c, whereby sinking POM is split or broken down into smaller POM and DOM. The newly formed total DOM at depth is further partitioned into $DOM_{SL}$ and $DOM_R$ at the same 1,000:7 ratio that occurs in the surface ocean. The rate of POM splitting into DOM depends on the availability of dissolved oxygen and temperature. The three pathways of $DOM_R$ remineralization in MESMO 3 are carried forward in MESMO 3c: slow background decay, photodegradation, and hydrothermal vent circulation, but these characteristic timescales of decay are calibrated. MESMO 3c reproduces distributions and inventories of total dissolved organic carbon ($DOC_T$) that are broadly consistent with observationally derived products.

In **MESMO 3 and MESMO 3c**, newly produced DOM—whether generated from NPP or through deep POM splitting—is partitioned into semi-labile and recalcitrant fractions using fixed allocation ratios (e.g., $fDOM_r$: ~1% or $DOM_{SL}$: $DOM_R$ = 1000:7). As a result, $DOM_R$ is designated as recalcitrant at the moment of production, rather than emerging through subsequent transformation.

By contrast, the **Microbial Carbon Pump (MCP)** framework emphasizes

the progressive reworking of labile and semi-labile DOC into recalcitrant compounds that accumulate over long timescales (Jiao et al., 2010; Jiao et al., 2024; Legendre et al., 2015). In **cGENIE-MCP**, the transformation from SLDOC to RDOC is implemented as an explicit, process-based pathway that is dynamically coupled to remineralization fluxes. Although a constant yield is prescribed, RDOC production depends on the time-evolving processing of semi-labile DOC. RDOC accumulation in cGENIE-MCP emerges from the time-integrated transformation of semi-labile DOC. The parameters governing this semi-labile–to–refractory DOC conversion are adopted from the observational data-constrained inverse modeling framework of Wang et al. (2023). These parameters were optimized using a Bayesian inversion approach that jointly assimilates global observations, yielding a model state that reproduces the observed large-scale DOC distribution with high fidelity. We have incorporated this observation-based, inverse-derived DOC transformation parameter into cGENIE-MCP. This process-based and data-informed representation distinguishes cGENIE-MCP from MESMO 3/3c schemes.

We note that MESMO 3c includes several DOC-related processes that are not yet represented in cGENIE-MCP, such as DOC degradation in hydrothermal vent systems. These differences reflect complementary modeling objectives. Our study is specifically designed to isolate and quantify the role of labile DOC transformation pathways emphasized by MCP theory. Additionally, the data and descriptions regarding cGENIE-MCP and MESMO3 in the manuscript have also been replaced with the updated version of MESMO3c. The main modifications in the revised manuscript are as follows:

"Despite recent advances in global marine DOC modeling, the explicit representation of MCP processes responsible for RDOC production in cGENIE remains limited. The Minnesota Earth System Model for Ocean biogeochemistry (MESMO 3) represents an important development, explicitly resolving semi-labile and refractory DOC pools. MESMO 3, developed based on the GENIE-1 framework, represents RDOC production diagnostically as a fixed fraction of organic matter production or via deep particulate organic matter (POM) partitioning, with RDOC removal governed by additive sink terms including slow background decay, surface photodegradation, and hydrothermal vent circulation (Matsumoto et al., 2021). Subsequent developments in MESMO 3c further refined this formulation by recalibrating DOC production relative to net primary production, introducing environmental dependencies such as temperature-dependent degradation rates, and constraining parameter values using global DOC observations (Gilchrist and Matsumoto, 2023). These refinements substantially improved agreement with observed DOC inventories and spatial patterns and represent an important advance in the simulation of large-scale DOC distributions. However, in both MESMO 3 and MESMO 3c, the assignment of organic matter to refractory DOC occurs at the point of production through prescribed allocation ratios, rather than emerging through an explicit representation of MCP-driven RDOC accumulation arising from the

progressive transformation of more labile DOC pools.

Here, we introduce cGENIE-MCP, an extension of the cGENIE model that explicitly represents MCP-driven DOC transformations. The framework partitions total DOC into three fractions—labile (LDOC), semi-labile (SLDOC), and refractory (RDOC)—and implements a process-based conversion of SLDOC into RDOC that is directly coupled to the remineralization process. In this formulation, RDOC accumulation emerges dynamically as a function of SLDOC remineralization processing rates and ocean circulation. We evaluate the performance of cGENIE-MCP against global observational datasets and compare its behavior with that of the standard cGENIE configuration. Finally, we analyze the spatial distribution and production of LDOC, SLDOC, and RDOC in relation to primary production to assess the model's ability to capture essential features of the MCP."

'a is a dimensionless conversion coefficient that represents the transformation of SLDOC into RDOC. The parameters governing the conversion from SLDOC to RDOC are derived from the observation-constrained inverse modeling framework of Wang et al. (2023). '

'4.3 Model performance for DOC

The statistical evaluation indicates that cGENIE-MCP reproduces observed DOC distributions with skill comparable to that of the well-established MESMO3c model across the major ocean basins (Table 32 and Fig. S12). The cGENIE-MCP yields low CRMSE in the Atlantic (3.63 µmol kg$^{-1}$), Pacific (3.84 µmol kg$^{-1}$), and Indian (2.56 µmol kg$^{-1}$) Oceans, MESMO3c shows similarly good performance in the Atlantic and Indian Oceans (Atlantic 4.03, Pacific 8.77, Indian 3.80 µmol kg$^{-1}$). When errors are weighted by model grid cell volumes (RMSE_vw), cGENIE-MCP achieves realistic basin-integrated DOC concentrations, with volume-weighted RMSE values ranging from 4-5 µmol kg$^{-1}$ across all major ocean basins. These results indicate that cGENIE-MCP provides a plausible representation of DOC when the volumetric contribution of different ocean layers is taken into account. Among all basins, the Indian Ocean shows the best performance for cGENIE-MCP, characterized by the lowest CRMSE and RMSE_vw values, possibly reflecting the model's enhanced representation of low-latitude processes. Taylor diagrams shows show the bias of modeled DOC from the MESMO3c and cGENIE-MCP models with observations (Figure 10). cGENIE-MCP exhibits a relatively high correlation coefficient and a smaller standard deviation comparable to the observed value in the Atlantic, Pacific and Indian Oceans.

[Figure]

Figure 10. Taylor diagrams comparing simulated DOC concentration from cGENIE-MCP and MESMO3c against observed values from Hansell's laboratory (https://hansell-lab.earth.miami.edu/research/data-collection/) for (A) Atlantic, (B) Indian, and (C) Pacific.

Table 2. RMSE of modeled DOC for cGENIE-MCP and MESMO3 compared to observations

| Tracers | cGENIE-MCP | | | MESMO3c | | |
|---|---|---|---|---|---|---|
| | CRMSE | RMSE_vw | R | CRMSE | RMSE_vw | R |
| Atlantic DOC (μmol kg⁻¹) | 3.63 | 4.48 | 0.966 | 4.03 | 2.13 | 0.938 |
| Pacific DOC (μmol kg⁻¹) | 3.84 | 5.33 | 0.919 | 8.77 | 5.37 | 0.614 |
| Indian DOC (μmol kg⁻¹) | 2.56 | 4.16 | 0.968 | 3.80 | 2.39 | 0.990 |

[Figure]

Figure S12. Global distributions of surface (A-B) DOC, (C-D) RDOC concentration (µmol kg$^{-1}$), (A,C) the results of cGENIE-MCP, (B,D) the results of MESMO3c.
,

**Reviewer comment 2: Gilchrist & Matsumoto (2023) have even carried out a glacial DOC cycle study, a long-term study that the authors of this submission hope to do.**

**Response:**

We did not intend to imply that glacial-scale or long-term DOC cycle studies have not been conducted previously. Indeed, Gilchrist & Matsumoto (2023) have already presented an important and comprehensive investigation of the glacial DOC cycle, and we fully acknowledge their contribution. Our intention was to indicate that the cGENIE-MCP framework developed in this study provides a basis for future applications of long-term DOC cycle simulations that explicitly incorporate MCP-driven DOC transformations within the cGENIE modeling framework. The primary objective of the present study is to investigate the relationship between MCP and other carbon pumps in Snowball Earth periods or the future. We have therefore revised the manuscript, and the main modifications are as follows:

'Several "Snowball Earth" events occurred throughout geological history. According to the Snowball Earth hypothesis, the biogeochemical cycle and the PP have severely slowed down or even stagnated under global freezing conditions. However, previous studies have found PP and DOC reservoirs still exist during glaciations (Jiao et al., 2024a; Man et al., 2024). These results suggest that organic matter produced in the surface ocean may have been degraded into DOC or RDOC in the water column. Therefore, understanding the changes of the DOC pools during Snowball Earth periods is of great significance (Hoffman and Schrag, 2002; Hoffman et al., 2017; Sansjofre et al., 2011). Indeed, previous studies in the glacial DOC cycle of Gilchrist and Matsumoto (2023) have demonstrated the importance of DOC dynamics during glacial climates. Building on these advances, a mechanistic characterization of the storage, spatial distribution, and source-sink processes of MCP-driven RDOC pools, as well as a quantitative assessment of their interactions with other carbon pumps during "Snowball Earth" periods, remains limited. The cGENIE-MCP model proposed in this study provides a process-based framework to simulate MCP-driven RDOC production and its large-scale spatial distribution over geological timescales. It is possible to analyze the relationship between δ$^{13}$C negative excursion and carbon pumps in geological records on a global scale, quantify the efficiency of MCP, and evaluate the impact of ocean environmental changes on the distribution of DOC.

Furthermore, reducing emissions and enhancing carbon sinks have

become a global consensus in response to global warming, with ocean carbon sinks playing a vital role in achieving this goal. Previous studies have pointed out that both "Snowball Earth" events and glacial-interglacial cycles are not only driven by orbital forcing but are also influenced by the ocean carbon cycle (Hoffman et al., 2017; Jiao et al., 2024a). The global ocean DOC reservoir is estimated to contain approximately 700 Pg C (Hansell, 2013). Although this accounts for only about 40% of the regenerated DIC reservoir (~1700 Pg C), it nonetheless represents a significant and long-lived carbon pool in the ocean. Its importance lies in its connection to SLDOC through the MCP process, allowing RDOC to vary in response to physical and biogeochemical perturbations. This dynamic behavior underscores the critical role of MCP-driven RDOC formation in regulating long-term ocean carbon storage and climate feedbacks. Therefore, evaluating the efficiency of the MCP is crucial for understanding of long-term climate regulation. The cGENIE-MCP model provides a flexible, modular framework that is well suited for long-term climate research. For example, by coupling the model with Shared Socioeconomic Pathway (SSP) scenarios, the response of MCP to rising atmospheric CO2 concentration can be investigated to reveal the feedback between climate change and the ocean carbon cycle. Future the model can be used to assess the potential of ocean negative carbon emission technologies (e.g., ocean alkalinization enhancement) under different climate scenarios. By simulating alternative implementation pathways, the long-term environmental impacts of these technologies can be quantified, enabling a comprehensive evaluation of optimal deployment strategies for sustainable carbon sequestration.'

**Reviewer comment 3: MESMO is "derived from cGENIE" is incorrect**

**Response:**
We initially stated that "MESMO is derived from cGENIE" to imply that MESMO is based on GENIE. According to literature review, MESMO is actually based on GENIE and extended its BGC module. While cGENIE represents a carbon-centric version of GENIE. Since throughout the manuscript we have been referring to cGENIE, we wrote "cGENIE" has caused some ambiguity. We have revised the manuscript as follows:

'The Minnesota Earth System Model for Ocean biogeochemistry (MESMO 3) represents an important development, explicitly resolving semi-labile and refractory DOC pools. MESMO 3 represents RDOC production as a fixed fraction of organic matter production or via deep particulate organic matter (POM) partitioning,…'

Geosci. Model Dev., 1, 1–15, 2008
www.geosci-model-dev.net/1/1/2008/

[Figure]

**Geoscientific
Model Development**

**First description of the Minnesota Earth System Model for Ocean biogeochemistry (MESMO 1.0)**

K. Matsumoto[1], K. S. Tokos[1], A. R. Price[2], and S. J. Cox[2]

[1]Department of Geology and Geophysics, University of Minnesota, Minneapolis, USA
[2]School of Engineering Sciences, University of Southampton, Southampton, UK

Received: 21 February 2008 – Published in Geosci. Model Dev. Discuss.: 27 March 2008
Revised: 19 June 2008 – Accepted: 15 July 2008 – Published: 4 August 2008

**Abstract.** Here we describe the first version of the Minnesota Earth System Model for Ocean biogeochemistry (MESMO 1.0), an intermediate complexity model based on the Grid ENabled Integrated Earth system model (GENIE-1). As with GENIE-1, MESMO has a 3D dynamical ocean, energy-moisture balance atmosphere, dynamic and thermodynamic sea ice, and marine biogeochemistry. Main development goals of MESMO were to: (1) bring oceanic uptake of anthropogenic transient tracers within data constraints; (2) increase vertical resolution in the upper ocean to better represent near-surface biogeochemical processes; (3) calibrate the deep ocean ventilation with observed abundance of radiocarbon. We achieved all these goals through a combination of objective model optimization and subjective targeted tuning. An important new feature in MESMO that dramatically improved the uptake of CFC-11 and anthropogenic carbon is the depth dependent vertical diffusivity in the ocean, which is spatially uniform in GENIE-1. In MESMO, biological production occurs in the top two layers above the compensation depth of 100 m and is modified by additional parameters, for example, diagnosed mixed layer depth. In contrast, production in GENIE-1 occurs in a single layer with thickness of 175 m. These improvements make MESMO a well-calibrated model of intermediate complexity suitable for investigations of the global marine carbon cycle requiring long integration time.

**1 Introduction**

Earth system Models of Intermediate Complexity (EMICs) occupy a unique and important position within the hierarchy of climate models (Claussen et al., 2002). In many ways, EMICs represent a compromise between high resolution, comprehensive coupled models of atmospheric and oceanic circulation, which require significant computational resources, and conceptual (box) models, which are computationally very efficient but represent the climate system in a highly idealized manner. A critical difference between comprehensive coupled models and box models is the absence of dynamical feedbacks in the latter. In box models, large scale circulation is typically prescribed and not allowed to change over the course of a simulation. The lack of dynamical feedbacks makes box models unsuitable for realistic simulations of transient climate change. On the other hand, comprehensive coupled models are so computationally intensive that their behavior within a given parameter space is difficult to fully explore. EMICs nicely fill this gap by retaining important dynamics while remaining computationally efficient, which is typically achieved by reducing spatial resolution and/or number of processes compared to high resolution coupled models.

The effectiveness of EMICs is evident in the numerous publications that have successfully employed them in studying past, present, and future climates (Ganopolski and Rahmstorf, 2001; Ganopolski et al., 1998; Joos et al., 1999; Knutti et al., 2002; Nusbaumer and Matsumoto, 2008; Plattner et al., 2001). Also, the important role that EMICs played in understanding the postindustrial carbon cycle changes is highlighted in the two recent IPCC science reports TAR (Houghton et al., 2001) and AR4 (IPCC, 2007).

Here we document development of the first version of the Minnesota Earth System Model for Ocean biogeochemistry (MESMO 1.0) based on an existing and successful EMIC called GENIE-1. Our immediate motivation for this work is to possess a tool to investigate postindustrial changes in the natural ocean carbon cycle. Our efforts were thus geared toward improving representation of marine biogeochemistry and distributions of natural and anthropogenic transient tracers in the oceans. These improvements, combined with a

[Figure]

*Correspondence to:* K. Matsumoto
(katsumi@umn.edu)

Fig. S1 The reference for MESMO1

**4 Description of MESMO**

The starting point of our model development is Version 6 of CB-GOLDSTEIN (Ridgwell et al., 2007), the non-modular version of GENIE-1. MESMO is identical to Version 6 unless noted otherwise. We decided not to use the word "GENIE" in our model name so as to avoid confusion with the ongoing efforts of the GENIEfy project to develop various flavors of GENIE. The GENIEfy project uses its SVN-controlled code and aims to modularize the different model modules, neither of which applies to our efforts with MESMO. The Bern 3D ocean model is also derived from C-GOLDSTEIN and also does not have the descriptor "GE-NIE" (Muller et al., 2006).

We describe MESMO's physical climate model (Sect. 4.1) first, followed by its biogeochemistry model (Sect. 4.2). In addition to describing the new features and modifications we adopted in MESMO, we will also briefly note two features that we evaluated but ultimately discarded (Sect. 4.3). Our dead-end efforts may be of some interest in future development efforts by other groups.

**4.1 New features in MESMO physical model**

First, the vertical resolution in the ocean is increased from 8 layers to 16. To allow biological production to depend on changes in stratification, it is preferable to have at least two layers in the euphotic zone above the critical depth where net production is positive. Therefore, we chose a vertical resolution that contains two complete layers in the top 100 m, which we took as the compensation depth (see Sect. 4.2 below). The midpoints of the 16 layers are: 23, 72, 133, 208, 300, 412, 550, 720, 927, 1182, 1494, 1877, 2347, 2923, 3630, and 4497 m. The increased vertical resolution is concentrated in the upper ocean such that the bottom topography in MESMO is very similar to GENIE-1, as shown in Fig. 1 of Ridgwell et al. (2007).
* * *
**Reviewer comment 4: Incorrect citation of discussion papers instead of final publications**

The reviewer notes that MESMO 3 and Lauvset et al. are cited as discussion papers rather than their final published versions.

Response:

We have re-examined all relevant citations and would like to clarify the following points. The citation to MESMO 3 in the original manuscript refers to the final published version, rather than the discussion paper. For MESMO 1, both the discussion paper and the final published article were cited simultaneously, rather than the discussion paper being cited alone.

Nevertheless, we acknowledge that citing both versions may lead to ambiguity. To avoid any potential confusion, we have removed the discussion paper citation for MESMO 1 and retained only the final peer-reviewed publication. In addition, we have carefully reviewed all references in the manuscript, including Lauvset et al., to ensure that only final published versions are cited and that all references conform to the journal's citation standards.

Despite recent advances, the explicit representation of MCP processes and RDOC cycling within cGENIE has remained limited. The Minnesota Earth System Model for Ocean biogeochemistry (MESMO 3), an Earth system model of intermediate complexity derived from cGENIE, includes an explicit treatment of semi-labile and refractory DOM pools. In MESMO3, the remineralization of refractory DOM

80   is represented by three additive sinks: slow background decay, surface photodegradation, and complete removal through hydrothermal vent circulation (Matsumoto et al., 2021). However, MESMO 3 lacks a mechanistic representation of DOM production pathways associated with MCP processes—specifically, the transformation of semi-labile DOC (SLDOC) into RDOC—and has not been calibrated against global DOM observations. In this study, we propose an extension to the cGENIE model that integrates RDOC

85   and MCP processes to investigate their long-term response to climate change. We introduce a new framework, cGENIE-MCP, which partitions total DOC into three distinct fractions—labile (LDOC), semi-labile (SLDOC), and refractory (RDOC)—enabling improved simulation of DOC production and remineralization based on prior formulations from cGENIE and MESMO 3. We evaluate the performance of the cGENIE-MCP model against global observational datasets and compare its outputs with those from

90   the standard cGENIE model. Finally, we analyze the spatial distribution and production of LDOC, SLDOC, and RDOC in relation to primary production to assess the model's capability in capturing essential features of the MCP.

Fig. S2 The reference to MESMO3 in the original text

**2.2 Ocean biogeochemical module**

The ocean biogeochemistry in cGENIE is simulated using the BIOGEM module, which models nutrient-driven biological productivity and the cycling of carbon and associated elements (Van De Velde et al., 2021). BIOGEM includes representations of air-sea gas exchange, nutrient uptake by primary producers, and the remineralization of organic matter in the water column. Phytoplankton are not explicitly represented; instead, primary productivity is calculated diagnostically based on the availability of limiting nutrients (phosphate and dissolved iron), solar radiation, and temperature, following parameterizations from previous studies (Matsumoto et al., 2008a; Matsumoto et al., 2008b; Matsumoto et al., 2021; Crichton et al., 2021). Phosphate ($PO_4$) and other nutrients are converted to particulate organic matter (POM) in the euphotic zone according to the Redfield stoichiometry. POMs are exported from the

Fig. S3 The reference to MESMO1 in the original text

[Figure]

Fig. S4 The corresponding list in the references section

Matsumoto, K., Tanioka, T., and Zahn, J.: MESMO 3: Flexible phytoplankton stoichiometry and refractory dissolved organic matter, Geoscientific Model Development, 14, 2265-2288, https://doi.org/10.5194/gmd-14-2265-2021, 2021.
690   Matsumoto, K., Tokos, K., Price, A., and Cox, S.: First description of the Minnesota Earth System Model for ocean biogeochemistry (MESMO 1.0), Geoscientific Model Development, 1, 1-15, https://doi.org/10.5194/gmd-1-1-2008, 2008.

Fig. S5 The corresponding list in the revised references section

**Reviewer comment 5: The reference to CMIP6 sounds like a strawman argument, because as the authors noted, CMIP models are used in "near-term climate projections." It really doesn't matter whether these models have refractory DOC or not.**

Response:

We agree that CMIP-class models are optimized for near-term projections. Our reference to CMIP models was intended purely as contextual motivation, highlighting the continued role of EMICs in addressing long-timescale carbon cycle questions. We have made revisions to the manuscript, and the main modifications are as follows:

'The Coupled Model Intercomparison Project Phase 6 (CMIP6) of Earth

System Models has significantly advanced the representation of physical and biogeochemical processes; however, the MCP-driven transformation of labile DOC into recalcitrant DOC remains highly simplified or implicitly represented in most models (Doney et al., 2024; Séférian et al., 2020b; Li et al., 2019). Many ESMs simplify DOC into a single dynamic pool or as multiple pools without explicit differentiation of transformation pathways and timescales (Anderson et al., 2015; Polimene et al., 2018; Ma et al., 2022; Flanjak et al., 2025). These structural simplifications mask the process-based role of MCP in progressively decreasing DOC lability and driving RDOC accumulation over decadal to millennial timescales, leading to underestimation of deep-ocean DOC concentrations and a failure to reproduce the millennial-scale radiocarbon ages observed in deep waters (Yamashita and Tanoue, 2008; Hansell et al., 2012; Follett et al., 2014). While introducing an explicit RDOC pool can improve simulated DOC concentrations and radiocarbon signatures, mechanistic MCP-based formulations provide additional insight into the processes governing RDOC accumulation and persistence (Hansell et al., 2012; Séférian et al., 2020a). '

**Reviewer comment 6: Equations are not labeled**

**Response:**
We have already labeled all the formulas in the manuscript.

**Reviewer comment 7: Unclear distinction between new developments and legacy code. Section 2.2.1 (air–sea gas exchange) appears unnecessary**

**Response:**
In the revised manuscript, the description of air-sea gas exchange has been moved to the Supporting Information.

Since there was no RDOM (corresponding to the code's URDOM) process in the cGENIE model code, the parts related to RDOM in the code were all newly added by us. The explicit partitioning of DOC into LDOC, SLDOC, and RDOC, and the process-based transformation of SLDOC into RDOC pools have also been added. Therefore, we have included the processes involving RDOM. We have also revised Section 2.2 to explicitly state which components of the biogeochemical model follow the legacy BIOGEM formulation and which aspects are newly developed.

The main modifications in the revised manuscript are as follows:
'Temperature-dependent nutrient uptake process of cGENIE in each

surface grid cell is carried forward in cGENIE-MCP and given by'

'The remineralization of POC is modeled as a temperature-dependent process of cGENIE is carried forward in cGENIE-MCP, with separate treatment for labile (POC1) and recalcitrant (POC2) components. The remineralization rate is given by:'

'LDOC is rapidly remineralized in the water column, releasing inorganic carbon and nutrients. The remineralization follows a similar temperature-dependent formulation of cGENIE is carried forward in cGENIE-MCP:'

**Reviewer comment 8: Table 1 is not referenced in the text**
**Response:**

We note that Table 1 was cited in the main text at two locations: first at Line 165, where we state "Key parameter values are given in Table 1", and again at Line 246, where we specify that "$f_1$, $f_2$, and a are listed in Table 1."

We have revised the surrounding text to more clearly: 'Key parameter values used to define the DOC cycling and MCP-related processes in this study are summarized in Table 1. '
'All other parameters are defined in the preceding equations, with the corresponding parameter values ($f_1$, $f_2$, and a) provided in Table 1.'

**Reviewer comment 11: Table 2 is not useful. It seems to be a global comparison, but the deep ocean is not highly variable. A global comparison would be biased toward the deep (i.e., global mean) just because of its large volume. Surface and intermediate depth comparisons would be more useful. And why does Table 2 include temperature and salinity? As far as I can tell, cGENIE-MCP has the same model physics as cGENIE.**

**Response:**

We have revised Table 2 and have moved it to the Supporting Information. We agree that a single global metric can be dominated by the large volume of the deep ocean and may obscure model-data differences in the upper and intermediate ocean. Therefore, the surface (0-100 m) and intermediate (100-1000 m) layers comparisons were added. For each tracer, we report both volume-weighted RMSE (RMSE_vw) and centered RMSE (CRMSE) within these depth ranges.

The temperature is included in Table 2 because the cGENIE-MCP configuration involves some temperature-related processes. Salinity is included primarily as a companion physical diagnostic. Although cGENIE-MCP employs the same physical circulation framework as standard cGENIE, this tracer together characterizes the model's water-mass structure and stratification,

which indirectly influence biogeochemical tracer distributions through circulation and mixing. These tracers are not included to demonstrate improvements introduced by the MCP formulation, but rather to verify that the introduction of MCP-driven DOC cycling does not introduce unintended degradation of the physical state of the model.

**Table 2. RMSE of modeled tracers for cGENIE-MCP and cGENIE and MESMO3c compared to observational data**

| Tracers | | cGENIE-MCP | | cGENIE | | MESMO3c | |
|---|---|---|---|---|---|---|---|
| | | CRMSE | RMSE_vw | CRMSE | RMSE_vw | CRMSE | RMSE_vw |
| T (°C) | 0-100m | 0.00 | 0.97 | 0.00 | 0.98 | 0.00 | 0.50 |
| | 100-1000m | 1.27 | 1.29 | 1.27 | 1.29 | 0.87 | 1.11 |
| Salinity | 0-100m | 0.00 | 0.07 | 0.00 | 0.08 | 0.00 | 0.13 |
| | 100-1000m | 0.12 | 0.16 | 0.12 | 0.16 | 0.13 | 0.14 |
| $PO_4$ (μmol kg$^{-1}$) | 0-100m | 0.00 | 0.19 | 0.00 | 0.51 | 0.00 | 0.03 |
| | 100-1000m | 0.13 | 0.37 | 0.10 | 0.56 | 0.10 | 0.36 |
| DO (μmol kg$^{-1}$) | 0-100m | 0.00 | 2.87 | 0.00 | 2.79 | 0.00 | 3.81 |
| | 100-1000m | 4.32 | 7.35 | 3.88 | 12.57 | 9.87 | 20.31 |
| DIC (μmol kg$^{-1}$) | 0-100m | 0.00 | 18.04 | 0.00 | 18.79 | 0.00 | 47.82 |
| | 100-1000m | 52.84 | 12.19 | 50.22 | 13.17 | 53.70 | 13.70 |

Jiao, N., Herndl, G. J., Hansell, D. A., Benner, R., Kattner, G., Wilhelm, S. W., Kirchman, D. L., Weinbauer, M. G., Luo, T., and Chen, F.: Microbial production of recalcitrant dissolved organic matter: long-term carbon storage in the global ocean, Nature Reviews Microbiology, 8, 593-599, https://doi.org/10.1038/nrmicro2386, 2010.

Jiao, N., Luo, T., Chen, Q., Zhao, Z., Xiao, X., Liu, J., Jian, Z., Xie, S., Thomas, H., and Herndl, G. J.: The microbial carbon pump and climate change, Nature reviews microbiology, 22, 408-419, https://doi.org/10.1038/s41579-024-01018-0, 2024.

Legendre, L., Rivkin, R. B., Weinbauer, M. G., Guidi, L., and Uitz, J.: The microbial carbon pump concept: Potential biogeochemical significance in the globally changing ocean, Progress in

Oceanography, 134, 432-450, https://doi.org/10.1016/j.pocean.2015.01.008, 2015.

Wang, W. L., Fu, W., Le Moigne, F. A. C., Letscher, R. T., Liu, Y., Tang, J. M., and Primeau, F. W.: Biological carbon pump estimate based on multidecadal hydrographic data, Nature, 624, 579-585, https://doi.org/10.1038/s41586-023-06772-4, 2023.